# Aurora kinase A inhibition reverses the Warburg effect and elicits unique metabolic vulnerabilities in glioblastoma

Trang T. T. Nguyen [1], Enyuan Shang[2], Chang Shu[1], Sungsoo Kim[1], Angeliki Mela[1], Nelson Humala[3], Aayushi Mahajan[3], Hee Won Yang [1], Hasan Orhan Akman[4], Catarina M. Quinzii[4], Guoan Zhang[5], Mike-Andrew Westhoff[6], Georg Karpel-Massler[7], Jeffrey N. Bruce [3], Peter Canoll[1] & Markus D. Siegelin [1✉]

Aurora kinase A (AURKA) has emerged as a drug target for glioblastoma (GBM). However, resistance to therapy remains a critical issue. By integration of transcriptome, chromatin immunoprecipitation sequencing (CHIP-seq), Assay for Transposase-Accessible Chromatin sequencing (ATAC-seq), proteomic and metabolite screening followed by carbon tracing and extracellular flux analyses we show that genetic and pharmacological AURKA inhibition elicits metabolic reprogramming mediated by inhibition of MYC targets and concomitant activation of Peroxisome Proliferator Activated Receptor Alpha (PPARA) signaling. While glycolysis is suppressed by AURKA inhibition, we note an increase in the oxygen consumption rate fueled by enhanced fatty acid oxidation (FAO), which was accompanied by an increase of Peroxisome proliferator-activated receptor gamma coactivator 1-alpha (PGC1α). Combining AURKA inhibitors with inhibitors of FAO extends overall survival in orthotopic GBM PDX models. Taken together, these data suggest that simultaneous targeting of oxidative metabolism and AURKAi might be a potential novel therapy against recalcitrant malignancies.

[1] Department of Pathology and Cell Biology, Columbia University Medical Center, New York, NY, USA. [2] Department of Biological Sciences, Bronx Community College, City University of New York, Bronx, New York, NY, USA. [3] Department of Neurological Surgery, Columbia University Medical Center, New York, NY, USA. [4] Department of Neurology, Columbia University Medical Center, New York, NY, USA. [5] Proteomics and Metabolomics Core Facility, Weill Cornell Medicine, New York, NY, USA. [6] Department of Pediatrics and Adolescent Medicine, Ulm University Medical Center, Ulm, Germany. [7] Department of Neurosurgery, Ulm University Medical Center, Ulm, Germany. ✉email: ms4169@cumc.columbia.edu

Glioblastoma WHO grade IV (GBM) is the most common primary brain tumor in adults and thus far very limited therapeutic options for this recalcitrant malignancy have been identified with essentially no durable response, resulting in a grim survival of only 12−15 months[1–6]. Key contributing factors of the resistance to therapy are the heterogeneity of these tumors, diffuse and infiltrative growth, the presence of the blood-brain barrier, and likely the plasticity of these tumors to reactivate alternate survival pathways. For the most part, glioblastomas consume glucose and utilize its carbons to facilitate biosynthesis of nucleotides and lipids to enable growth, while oxidative metabolism is relatively suppressed (Warburg-effect)[7,8]. The recent literature highlights a role for the transcription factor, c-Myc, which is known to facilitate several aspects of this pathway by driving the expression of key glycolytic and anabolic enzymes[9].

Aurora kinases are important for the proliferation and growth of solid tumors, including glioblastomas. They are phosphorylating several substrates that are directly involved in cell cycle regulation. Several members have been described and Aurora kinase A has been identified as a target for glioblastoma therapy, especially because specific inhibitors exist (alisertib, MLN8237) that have shown limited single-agent efficacy in orthotopic GBM model systems. Importantly, clinical trials have been initiated with these compounds as well[10–12]. For this reason, a better understanding about the response and resistance of these compounds is warranted to arrive at more efficacious treatments.

c-Myc (MYC) is an oncogenic transcription factor that facilitates tumor proliferation in part through the regulation of metabolism. It carries a short half-life, and its stability is regulated by two interdependent phosphorylation sites, Serine 62 (S62) and Threonine 58 (T58). The phosphorylation at S62, which is facilitated by ERK, precedes the one at T58, which is mediated by GSK3β and subjects c-Myc to ubiquitin conjugation via the ubiquitin ligase FBXW7[13].

In this study, we investigated how Aurora kinase A inhibition affects glioblastoma cell metabolism and how the derived knowledge from these studies can be leveraged for the design of drug combination therapies. We made the unexpected and important finding that the response to Aurora kinase A inhibitors depends on glycolysis and that tumor cells with an oxidative metabolic phenotype will be more resistant to Aurora kinase A inhibitor treatment. Moreover, in a manner dependent on the transcription factors c-MYC and PGC1α treatment with Aurora kinase A inhibitors renders GBM cells highly oxidative and dependent on fatty acid oxidation that in turn mediates them to be susceptible to inhibitors of FAO in vitro and in vivo.

## Results

### Aurora kinase A regulates proliferation and glycolysis through the oncogene, c-Myc.
Following proteomic analysis, we found that SF188 GBM cells treated with alisertib, a clinically validated highly specific AURKA inhibitor, displayed substantial downregulation of the c-Myc protein (Fig. 1a), a master regulator of cell proliferation and metabolism[9]. We chose c-Myc over other targets due to its substantial role in tumorigenesis and its implication in aerobic glycolysis. A complete list of all the targets is provided in the source data file. Subsequently, we confirmed this observation by performing protein capillary electrophoresis in SF188 and GBM22 PDX cells (Fig. 1b). Treatment with alisertib abolished autophosphorylation of Aurora kinase A at Thr288 (T288) accompanied by a decrease in c-Myc protein levels (Fig. 1b and Fig. S1a). Consistently, alisertib reduced c-Myc protein levels in a dose-dependent manner as early as 1 h in these model systems (Fig. S1b, c). We also confirmed that alisertib revealed anti-glioma activity in the GBM22 orthotopic PDX

model, revealing smaller tumors following treatment with the Aurora kinase A inhibitor (Fig. S1d, e). In addition, we also noted that c-Myc protein levels substantially decreased following treatment with alisertib in an orthotopic murine GBM model (Fig. S1f–i). Consistently, we found reduced phosphorylation of Aurora kinase A following treatment with alisertib (Fig. S1f, g). Next, we genetically targeted AURKA by using siRNA, shRNA, or CRISPR/Cas9 in GBM cells (Fig. 1c). Resembling the alisertib treatment, we noted a reduction in c-Myc protein levels in AURKA knock down and knock out cells. To confirm that c-Myc is the actual effector of Aurora kinase A-mediated inhibition of proliferation, we silenced c-Myc by two specific siRNA. Silencing of c-Myc resulted in a significant reduction of the effect of AURKA inhibition on cellular viability in SF188 and GBM22 cells (Fig. 1d), suggesting that c-Myc is a key mediator in this context. Similarly, when c-Myc was silenced alisertib lost its ability to further elicit induction of apoptosis in GBM cells (Fig. S2a, b). Consistently, dual silencing of c-Myc and AURKA did neither yield further enhancement of apoptosis nor reduction in cellular viability as compared to silencing of each gene on its own (Fig. S2c−e). Next, we determined whether over-expression of c-Myc could partially attenuate the effects of alisertib-mediated reduction in cellular viability in GBM cells. To this end, we utilized an adenovirus encoding for c-Myc. Indeed, SF188 and GBM22 cells that over-expressed c-Myc were slightly less sensitive to alisertib mediated reduction in viability (Fig. 1e). In summary, these results confirm the notion that loss of c-Myc is functionally involved in the reduction of viability elicited by Aurora kinase A inhibition.

Next, we investigated how Aurora kinase A inhibition leads to a reduction in c-Myc protein levels. The c-Myc protein levels are known to be heavily regulated by the proteasome[9,14–16]. Hence, treatment with alisertib did not reduce c-Myc mRNA levels (Fig. S2f). Next, we assessed whether Aurora kinase A interacts with c-Myc. To this end, we performed co-immunoprecipitation analysis and found that c-Myc bound to Aurora kinase A in both SF188 and GBM22 cells (Fig. 1f). This was also confirmed in 293T cells by co-transfection of Flag-tagged c-Myc and HA-tagged AURKA (Fig. S3a). Similarly, proximity ligation assay demonstrated an interaction between c-Myc and Aurora kinase A in both SF188 and GBM22 cells (Fig. S3b−e). Next, we assessed whether alisertib was capable to disrupt the interaction between c-Myc and Aurora kinase A. To this purpose, SF188 GBM cells were treated with increasing dosages of alisertib, followed by c-Myc immunoprecipitation. While the vehicle-treated cells showed a strong interaction between c-Myc and Aurora kinase A, higher concentrations of the drug elicited a disruption of the complex (Fig. S3f, g). Furthermore, we probed these lysates for GSK3β. We found that the relative binding ratio between GSK3β and Aurora kinase A to c-Myc increased following alisertib treatment in a concentration-dependent manner, suggesting that alisertib favors the binding of GSK3β to c-Myc over Aurora kinase A, in keeping with the overall reduction of c-Myc protein levels (Fig. S3f, g). Akin to the co-IP assay, the proximity ligation assay confirmed that alisertib disrupted the interaction between Aurora kinase A and c-Myc (Fig. S3h, i). Given that direct silencing of Aurora kinase A leads to a suppression of c-Myc protein levels, it appears likely that Aurora kinase A acts as a chaperone to protect c-Myc from proteasomal degradation. To address this hypothesis, we utilized a two-pronged strategy. First, we treated SF188 cells with alisertib in the presence or absence of the proteasomal inhibitor, MG132. As expected, we found that MG132 substantially reversed alisertib-mediated suppression of c-Myc (Fig. 1g). Second, we conducted a cycloheximide block experiment. Consistently, the stability of c-Myc was significantly reduced in the presence of alisertib (Fig. 1h, i and Fig. S3j). These

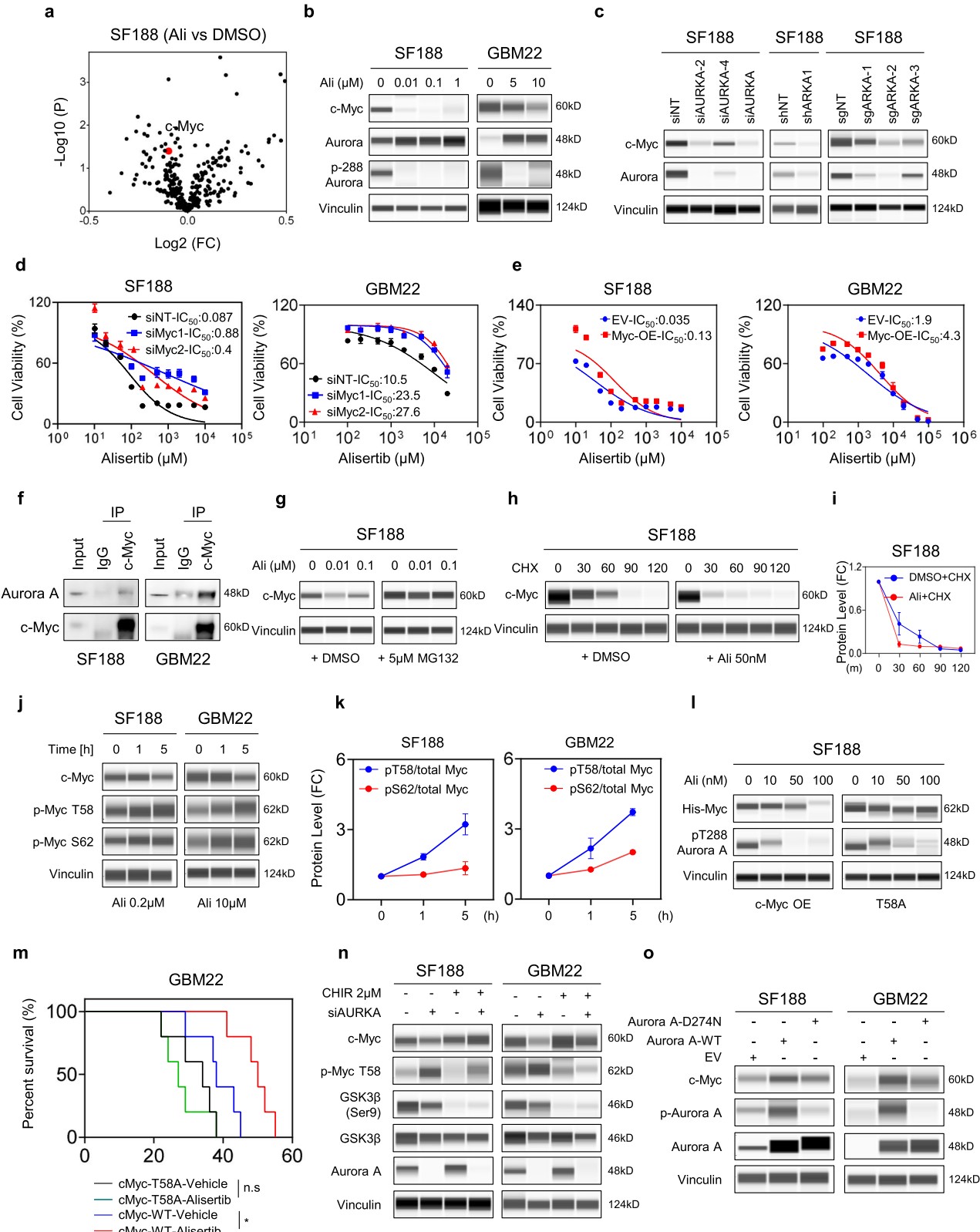

two observations narrow the range and strongly favor a mechanism that involves the proteasome following Aurora kinase A inhibition. Classically, c-Myc is phosphorylated prior to its degradation and there are two sites that have been reported to orchestrate this, S62 and T58[15,16]. The phosphorylation at S62 precedes the one at T58, which is mediated by GSK3β and subjects c-Myc to ubiquitin conjugation. In agreement with this general concept, alisertib enhanced the phosphorylation at S62 and T58, respectively, which occurred as early as one hour following exposure to the drug compound, indicating that these two posttranslational modifications are early events that precede c-Myc degradation (Fig. 1j, k). To further elucidate the role of these posttranslational modifications we utilized a T58A non-phosphorylatable c-Myc mutant. Ectopic expression of the T58A

**Fig. 1 Blocking AURKA stalls the proliferation of GBM by reducing glycolysis in a c-Myc dependent manner. a** Volcano plot of reverse-phase-protein-array (RPPA) data shows a reduced expression of c-Myc (red dot) in SF188 GBM cells treated with 100 nM alisertib for 24 h ($n = 3$ independent samples). FC: fold change. **b** Protein capillary electrophoresis of SF188 and GBM22 cells treated with the indicated concentrations of alisertib for 24 h. Vinculin is used as a loading control. **c** SF188 was transfected with non-targeting (siNT) or AURKA specific siRNA (siAURKA) or transduced with shARKA (shRNA) or sgAURKA (sgRNA) and the whole-cell protein lysates were subjected to protein capillary electrophoresis. **d** SF188 and GBM22 cells were transfected with non-targeting (siNT) or two specific siRNAs targeting Myc, treated with increasing concentrations of alisertib for 72 h, and cellular viability was analyzed ($n = 4$ independent samples). IC$_{50}$ in µM range. **e** SF188 and GBM22 cells were infected with empty vector or c-Myc adenovirus and treated with increasing concentration of alisertib for 72 h, and cellular viability was analyzed ($n = 4$ independent samples). IC$_{50}$ in µM range. **f** SF188 and GBM22 cell lysates were immunoprecipitated with IgG or c-Myc antibody and analyzed by standard western blot for the indicated antibodies. Input: cell lysate loading control. IgG: negative control. **g** SF188 cells were treated with DMSO or alisertib in the presence or absence of 5 µM MG132 and the whole-cell lysates were subjected to protein capillary electrophoresis for the indicated proteins. For **h**, **i** SF188 cells were treated with DMSO or alisertib in the presence or absence of 10 µg/mL cycloheximide (CHX) and the whole-cell lysates were subjected to protein capillary electrophoresis. Quantification of c-Myc protein level is shown in (**i**) ($n = 3$ independent samples). For **j**, **k** protein capillary electrophoresis for the indicated proteins of SF188 and GBM22 cells treated with DMSO or alisertib for different time points. Quantification of protein level is shown in (**k**) ($n = 2$ independent samples). **l** SF188 cells were transfected with c-Myc-WT or c-Myc mutant (T58A), treated with the indicated concentrations of alisertib for 24 h, and analyzed by protein capillary electrophoresis for the indicated proteins. **m** GBM22-Myc-WT or GBM22-T58A-Myc cells were implanted in the right striatum of nude mice. Two groups were randomly assigned: vehicle and alisertib after seven days of the implantation. Mice were treated three times per week and animal survival is provided (Kaplan−Meier-curve): GBM22-T58A-Myc-vehicle: 34d, GBM22-T58A-Myc-alisertib: 27d; GBM22-Myc-OE-vehicle: 38d, GBM22-Myc-OE-alisertib: 50d. The log-rank test was used to assess statistical significance ($n = 5$ independent samples) (*$p = 0.0127$, n.s not significant). **n** SF188 and GBM22 cells were transfected with non-targeting siNT or siAURKA in the presence or absence of 2 µM CHIR-908014 (CHIR) for 24 h and were subjected to protein capillary electrophoresis for the indicated proteins. **o** SF188 and GBM22 cells were transfected with HA-EV, HA-Aurora A-WT, HA-Aurora A-D274N and were subjected to protein capillary electrophoresis for the indicated proteins. Statistical significance was assessed by two-tailed student's *t*-test in (**m**). Data are shown as mean ± SD in (**d**, **e**, **i**, **k**). Source data are provided as a Source Data file.

c-Myc mutant rendered GBM cells resistant towards alisertib-mediated c-Myc degradation (Fig. 1l and Fig. S3k). The substantial impact on c-Myc stability conferred by the phospho-mutant (T58A) led us to hypothesize that c-Myc likely has a central role in mediating the response and resistance to alisertib. To this end, we transduced GBM22 cells with lentiviral particles encoding c-Myc T58A and c-Myc wild-type (Fig. S3l). Stably transduced GBM22 cells were implanted in the right striatum. Upon tumor formation, animals were either treated with vehicle or alisertib until the animals met the endpoint. As anticipated, host animals carrying orthotopically implanted GBM22 cells expressing the T58A mutated c-Myc were resistant towards alisertib treatment, whereas animals carrying wild-type c-Myc orthotopic xenografts were sensitive to the drug and revealed an extended overall survival (Fig. 1m).

To confirm the involvement of GSK3β in c-Myc-mediated degradation, we interfered with the function of this kinase. Indeed, the GSK3β blocker, CHIR-908014, prevented alisertib-mediated degradation of c-Myc (Fig. 1n and Fig. S4a). In alignment with this finding, silencing of AURKA or pharmacological inhibition reduced the phosphorylation of GSK3β, rendering this kinase more active, resulting in enhanced phosphorylation and degradation of c-Myc (Fig. 1n and Fig. S4a, b). In addition to pharmacological inhibition of GSK3β, we silenced this gene, using siRNA. In line with the pharmacological inhibition, genetic interference with GSK3β reduced alisertib-mediated reduction of c-Myc levels (Fig. S4c, d). We wondered whether AURKA would bind to GSK3β to block its function. Indeed, co-immunoprecipitation studies indicated that GSK3β and Aurora kinase A formed a complex in SF188 GBM cells (Fig. S4e). While a direct binding between these proteins has been shown earlier[17], the interaction of the two kinases in GBM models and its impact on c-Myc are unique. Further confirmation of the GSK3β and Aurora kinase A complex was obtained by utilizing the proximity ligation assay, which confirmed the findings in both GBM22 and SF188 cells (Fig. S4f−i). Next, we wondered whether AURKA could simultaneously interact with both c-Myc and GSK3β. To this end, we transfected SF188 and GBM22 cells with HA-tagged AURKA and performed co-IP. We found that indeed both c-Myc and GSK3β are bound to Aurora

kinase A (Fig. S5a). To further clarify this interaction and the role of Aurora kinase A we performed an additional co-IP analysis in which three plasmids (untagged-AURKA, Flag-GSK3β, and HA-c-MYC) were simultaneously transfected. The plasmid encoding for Aurora kinase A was used at three different dosages to demonstrate that Aurora kinase A disrupts the interaction between c-Myc and GSK3β and thereby inhibits the phosphorylation of c-Myc at T58 by GSK3β (Fig. S5b). To extent these findings related to MYC phosphorylation/degradation further in the context of glioblastoma cells, we have generated a kinase-dead mutant of Aurora kinase A (D274N). When compared to wild-type Aurora kinase A the kinase-dead mutant revealed reduced ability to phosphorylate GSK3β at serine 9 (Fig. S5c). Consistently, we observed that only the wild-type, but not the kinase-dead mutant Aurora kinase A was able to increase c-Myc protein levels in both SF188 and GBM22 cells (Fig. 1o). Thus, these observations support the idea that Aurora kinase A regulates the phosphorylation of c-Myc by GSK3β in a kinase-dependent manner. In summary, AURKA stabilizes the c-Myc protein and antagonizes GSK3β mediated phosphorylation of c-Myc and its subsequent degradation mediated by the proteasome.

**Aurora kinase A drives GBM proliferation through activation of glycolysis in GBM cells.** Based on a high throughput drug screen[18] we made the intriguing observation that interference with Aurora kinase A and the respiratory chain may be synthetically lethal, suggesting that blockage of Aurora kinase A signaling may activate oxidative metabolism and that this, in turn, may be used by GBM cells to acquire resistance. Conversely, while Aurora kinase A might activate respiration it likely would reduce glycolysis. Since c-Myc is a known master regulator of the Warburg-effect, controlling the expression of several key glycolytic genes, it was tempting to speculate whether Aurora kinase A inhibition would reduce the proliferation of GBM cells in a manner dependent on glycolysis, which is controlled by c-Myc. To determine whether alisertib reduces cellular viability through regulation of glycolysis GBM cells were grown in galactose-containing medium, which reduces glycolytic flux and forces cells to depend on mitochondrial oxidative phosphorylation for energy production (Leloir pathway). The GBM cells grown in glucose are

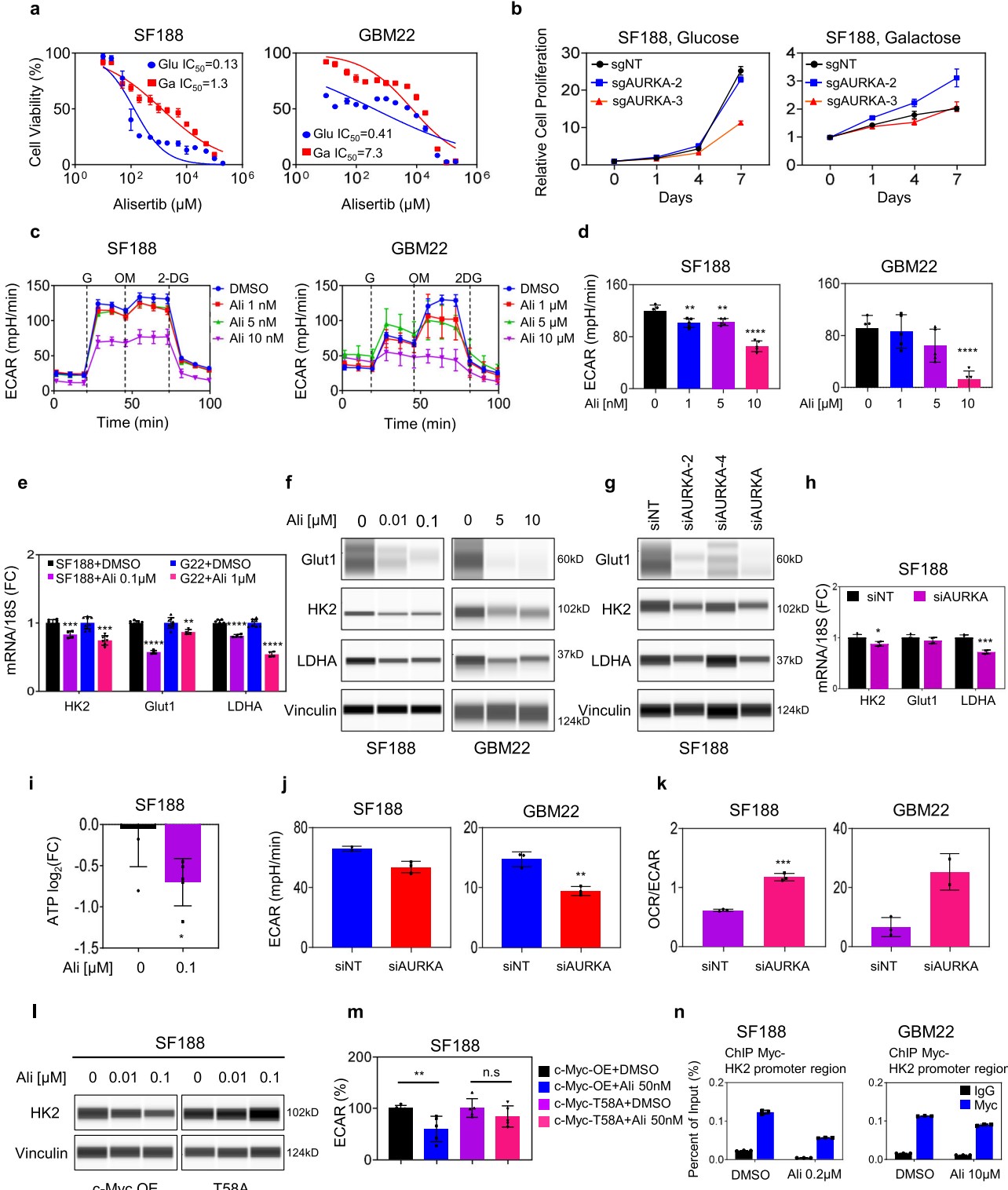

more sensitive to alisertib treatment or genetic interference (CRISPR/Cas9) with AURKA compared to the ones grown in galactose (Fig. 2a, b).

Next, we determined whether Aurora kinase A inhibition suppresses glycolysis. To this purpose, we used extracellular flux analysis (Seahorse analyzer) in the context of a glycolytic stress assay. Cells are initially starved, exposed to glucose followed by injection of oligomycin to exacerbate the highest glycolytic activity. The assay is completed with 2-DG to block glycolysis. In

this setting, we noted that alisertib suppressed the activity of glycolysis in both SF188 and GBM22 PDX cells (Fig. 2c, d). To account for specificity, we utilized an Aurora kinase A specific siRNA and conducted the glycolytic stress assay. Similarly, to the drug compound, silencing of Aurora kinase A diminished glycolytic activity, suggesting that Aurora kinase A regulates glycolysis in GBM model systems (Fig. S6a). Next, we determined the impact of Aurora kinase A on key glycolytic transporters (SLC2A1) and enzymes (HK2, LDHA), some of which are

**Fig. 2 Blocking AURKA stalls GBM growth by reducing glycolysis. a** SF188 and GBM22 cells were cultured in galactose or glucose media for a week, treated with increasing concentrations of alisertib for 72 h, and cellular viability was analyzed ($n = 4$ independent samples). **b** SF188 GBM cells either transduced with a non-targeting (sgNT) or two different AURKA (sgAURKA) sgRNAs were cultured in galactose or glucose media for a week and treated with increasing concentration of alisertib. Cellular growth over time was determined ($n = 4$ independent samples). For **c, d** SF188 and GBM22 cells were treated with DMSO or alisertib and analyzed in the context of a glycolysis stress assay on a Seahorse XFe24 extracellular flux analyzer. Extracellular acidification rate (ECAR) is recorded at baseline, after injection of Glucose (G), Oligomycin (OM), and 2-DG. Quantification of glycolysis is shown in (**d**) ($n = 5$ independent samples). **e** Real-time PCR analysis of the mRNA level of genes related to glycolysis in SF188 and GBM22 cells treated with DMSO or alisertib for 24 h ($n = 8$ in SF188 + DMSO and GBM22 + DMSO, $n = 4$ in SF188 + Ali 0.1 μM and GBM22 + Ali 1 μM, independent samples). (HK2: SF188 ***$p = 0.0002$, GBM22: ***$p = 0.0003$; **$p = 0.0082$, ****$p < 0.0001$). 18S: internal control. FC: fold change. **f** SF188 and GBM22 cells were treated with DMSO or alisertib for 24 h and the whole-cell protein lysates were subjected to protein capillary electrophoresis. Vinculin is used as a loading control. **g** SF188 cells were transfected with non-targeting siRNA or specific AURKA siRNAs (single or pool) and the whole-cell lysates were analyzed by protein capillary electrophoresis for the indicated proteins. **h** Real-time PCR analysis of the mRNA level of genes related to glycolysis in SF188 transfected with non-targeting siRNA or specific siRNA targeting AURKA ($n = 4$ independent samples) (*$p = 0.0179$, ***$p = 0.0001$). **i** Shown are the ATP levels measured by polar LC/MS of SF188 cells treated with DMSO or 100 nM alisertib ($n = 5$ independent samples) (*$p = 0.0221$). For **j, k** SF188 and GBM22 cells were transfected with non-targeting or specific siRNA targeting AURKA and were analyzed in the context of a glycolysis stress assay on a Seahorse XFe24 extracellular flux analyzer. The graphs show glycolysis level in (**j**) (SF188: $n = 2$ in siNT, $n = 3$ in siAURKA; GBM22: $n = 3$ independent samples) (**$p = 0.0032$) or OCR/ECAR levels in (**k**) (SF188: $n = 3$; GBM22: $n = 3$ in siNT, $n = 2$ in siAURKA independent samples) (***$p = 0.0001$). **l** SF188 cells were transfected with c-Myc-WT and c-Myc mutant (T58A), treated with increasing concentrations of alisertib for 24 h, and the whole-cell lysates were analyzed by protein capillary electrophoresis. **m** SF188 cells were transfected with c-Myc-WT and c-Myc mutant (T58A), treated with 50 nM alisertib for 24 h, and analyzed on a Seahorse XFe24 extracellular flux analyzer. Shown is the quantification of ECAR level ($n = 5$ independent samples) (**$p = 0.0073$, n.s not significant). **n** SF188 and GBM22 cells were treated with alisertib for 24 h and were subjected to CHIP with an IgG as a negative control or a c-Myc specific antibody. The HK2 region was amplified by PCR ($n = 3$ independent samples). Statistical significance was assessed by a two-tailed student's $t$-test. Data are shown as mean ± SD in (**a**−**e**, **h**−**k**, **m**, **n**). Source data are provided as a Source Data file.

upregulated in glioblastoma as compared to normal brain tissue (Fig. S6b). Following treatment with alisertib or specific silencing of Aurora kinase A we detected a significant suppression of both mRNA and protein levels of GLUT1, HK2, and LDHA in SF188 and GBM22 cells (Fig. 2e–h). We also confirmed the down-regulation of these transcripts in an orthotopic GBM PDX model that we used earlier for the confirmation of the suppression of c-Myc protein levels (Figs. S1h, l and S6c). While glycolysis was suppressed (accompanied by a reduction of ATP levels) we detected a concomitant increase in the oxygen consumption rate, indicating that activation of oxidative metabolism might partially counteract for the loss of glycolytic activity (Fig. 2i–k).

Given that genes related to glycolysis were suppressed at the mRNA level following interference with Aurora kinase A, we hypothesized that c-Myc may be the transcriptional regulator of these changes. Although it has been shown earlier that c-Myc can regulate the expression of HK2 and LDHA, we validated these findings in our studied model systems[9]. To this purpose, SF188 cells were transfected with c-Myc specific siRNA or non-targeting siRNA. Protein capillary electrophoresis demonstrated that silencing of c-Myc suppressed the expression of c-Myc and related glycolytic targets, SLC2A1, HK2, and LDHA (Fig. S6d). Next, we linked the effects of Aurora kinase A inhibition on HK2 directly with c-Myc through silencing and over-expression experiments. As anticipated, when c-Myc is silenced alisertib did not further reduce the expression levels of HK2 (Fig. S6e). Next, we intended to functionally connect the effects of Aurora kinase A on glycolysis via c-Myc. To this end, we performed silencing experiments followed by a glycolytic stress assay in GBM22 cells. Genetic interference with c-Myc and Aurora kinase A led to a suppression of glycolysis and related parameters (Fig. S6f). Notably, combined silencing of c-Myc and Aurora kinase A did not result in a further reduction of glycolysis, suggesting that the effect of Aurora kinase A on glycolysis is largely explained by c-Myc. Conversely, overexpression of the c-Myc T58A mutant in SF188 and GBM22 cells attenuate alisertib mediated suppression of HK2, suggesting that Aurora kinase A regulates the expression of HK2 via c-Myc (Fig. 2l and Fig. S6g). Next, we assessed whether over-expression of c-Myc would rescue from alisertib-mediated reduction in glycolysis. To this purpose, we transfected

c-Myc or T58A mutated c-Myc into SF188 and U87 GBM cells. Following transfection, cells were exposed to treatment with alisertib and subjected to extracellular flux analysis (glycolysis stress assay) (Fig. 2m and Fig. S6h). As anticipated forced expression of T58A c-Myc rescued from alisertib-mediated reduction in glycolysis. Next, we assessed whether there is also reduced binding of c-Myc to the promoter region of HK2. To this end, we conducted chromatin immunoprecipitation in the vehicle and acute and chronic treatments with alisertib. As anticipated, we found that acute and chronic alisertib exposure reduced the binding of c-Myc to the HK2 promoter in GBM cells, further confirming the proposed cascade (Fig. 2n and Fig. S6i).

**Acute and chronic Aurora kinase A inhibition activates oxidative energy metabolism.** While glycolysis was suppressed by AURKA inhibition, we noted a compensatory increase in the oxygen consumption rate (Fig. 2j, k). To extend this finding further, extracellular flux analysis (mitochondrial stress assay) was performed following acute and chronic inhibition of Aurora kinase A in several GBM cells. We detected an increase in oxygen consumption rate following acute AURKA inhibition with alisertib (Fig. 3a and Fig. S7a), while the same conditions yielded a suppression of glycolysis, indicative of a reversal of the "Warburg effect". Besides from the acute treatment, it is important to understand how tumor cells acquire mechanisms to escape from chemotherapy following constant exposure to a drug and identify means to prevent this phenomenon from occurring. To generate drug-resistant cells, they were cultured in the presence of alisertib for two weeks. These cells acquire partial resistance to alisertib and display a hyper-oxidative phenotype (Fig. 3b, c, and Fig. S7b, c). To validate the specificity of these findings, we assessed GBM cells with shRNA or CRISPR-mediated suppression of AURKA. As anticipated, the increase in oxygen consumption rate was also seen following genetic interference with AURKA (Fig. 3d and Fig. S7d). Coupled with this oxidative signature, electron microscopy revealed an increase in the size of mitochondria with a tubulated shape following interference with Aurora kinase A signaling (Fig. 3e). Consistently, acute alisertib treatment showed an increase in mitochondrial size as assessed by flow cytometric analysis (Fig. 3f).

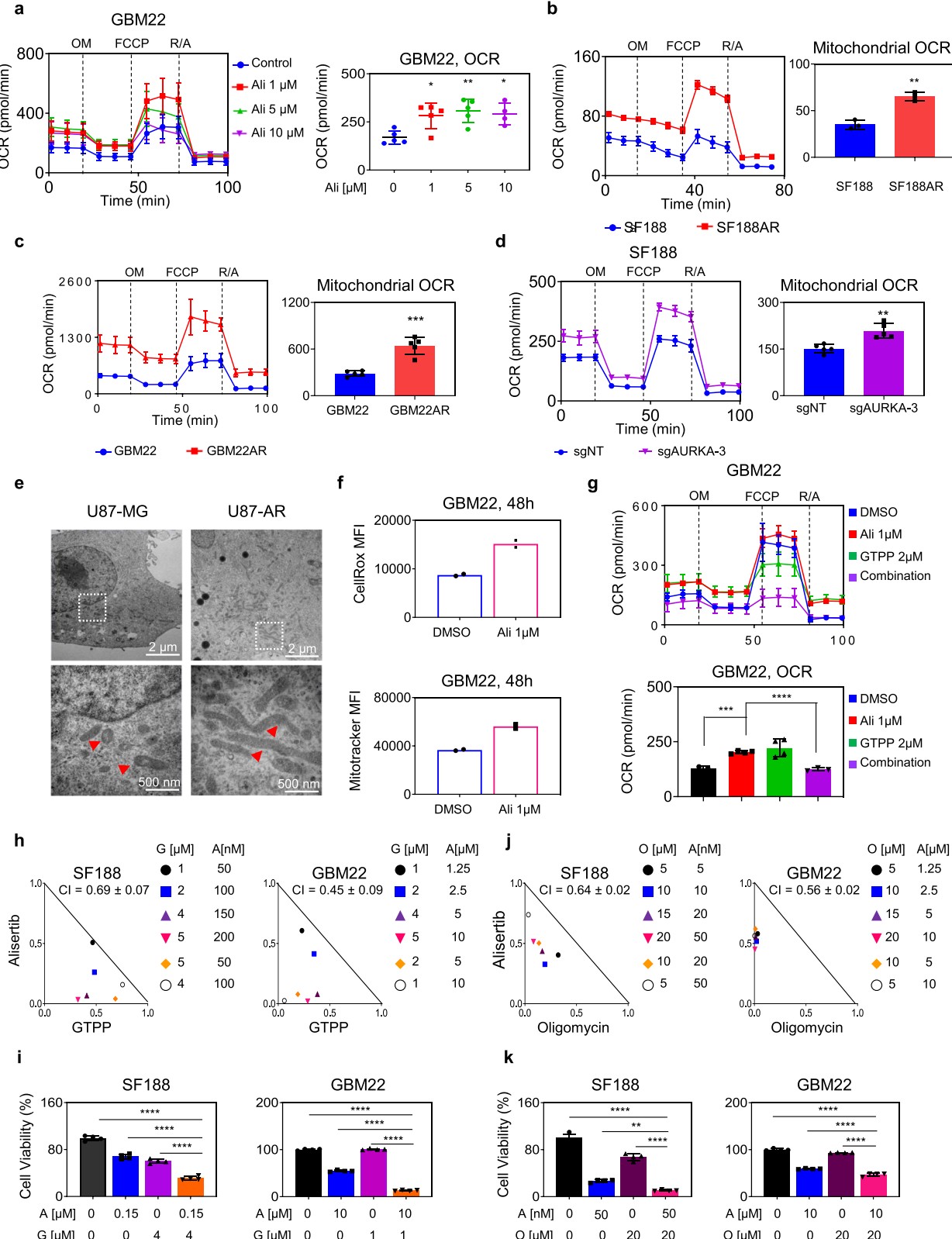

**Inhibition of Aurora kinase A and oxidative phosphorylation is synthetically lethal in GBM model systems and other solid malignancies**. To assess whether OXPHOS inhibition enhances the efficacy of AURKA inhibition mediated reduction in cellular viability, we used several OXPHOS complex inhibitors such as gamitrinib (GTPP, blocks complex II of the respiratory chain), metformin (blocks complex I of the respiratory chain), and

oligomycin (inhibits mitochondrial complex V (ATP synthase)). We launched our studies with gamitrinib to validate that indeed gamitrinib reversed alisertib mediated increase in oxidative metabolism. To this end, we utilized extracellular flux analysis following treatment with vehicle, alisertib, gamtrinib, or the combination (Fig. 3g). While alisertib increased the oxygen consumption rate, it was suppressed upon treatment with the

**Fig. 3 AURKA inhibition facilitates oxidative energy metabolism. a** GBM22 cells were treated with DMSO or alisertib and analyzed for oxygen consumption rate (OCR) on a Seahorse XFe24 device. OM: oligomycin, R/A rotenone/antimycin. The graph (right panel) shows the OCR levels ($n = 5$ in DMSO, Ali 1 μM, Ali 5 μM; $n = 4$ in Ali 10 μM independent samples) (Ali 1 μM: $*p = 0.0303$, Ali 5 μM: $**p = 0.0074$, Ali 10 μM: $*p = 0.0254$). For **b, c** Seahorse mitochondrial stress assay of parental or chronically alisertib treated GBM22 and SF188 cells. The graph (right panel) shows the mitochondrial OCR levels ($n = 3$ in SF188 vs SF188AR, $n = 5$ in GBM22 vs GBM22AR independent samples) ($**p = 0.0014$, $***p = 0.0001$). **d** SF188 cells were transduced with sgRNA against AURKA and were analyzed for the oxygen consumption rate (OCR) on a Seahorse XFe24 device. The graph (right panel) shows the mitochondrial OCR level ($n = 5$ independent samples) ($**p = 0.0016$). **e** Electron microscopy of parental or chronically alisertib treated U87 cells. Scale bar: 2 μm. Higher magnification images are shown in the lower panel. Scale bar: 500 nm. The white dotted square shows the enlarged region of mitochondria in the lower panel. The red arrow highlights the mitochondrial morphology. **f** GBM22 cells were treated with DMSO or alisertib for 24 h, labeled with either CellRox or Mitotracker dye, and analyzed by flow cytometry ($n = 2$ independent samples). **g** GBM22 cells were treated with 1 μM alisertib, 2 μM GTPP, or the combination of both for 24 h and analyzed for oxygen consumption rate (OCR) on a Seahorse XFe24 device. The graph (below panel) shows the mitochondrial OCR level ($n = 3$ in DMSO and Combination, $n = 4$ in Ali 1 μM and GTPP 2 μM independent samples) ($***p = 0.0001$, $****p < 0.0001$). For **h, i** SF188 and GBM22 cells were treated with alisertib, GTPP, or the combination of both for 72 h. Thereafter, cellular viability was measured, and statistical analysis was performed. Isobolograms are shown. The graphs in (**i**) show the quantification ($n = 4$ independent samples) ($****p < 0.0001$). For **j, k** SF188 and GBM22 cells were treated with alisertib, oligomycin, or the combination of both for 72 h. Thereafter, cellular viability was determined and statistical analysis was performed. Isobolograms are shown. The quantification is shown in (**k**) ($n = 4$ independent samples) ($**p = 0.0015$, $****p < 0.0001$). Data are shown as mean ± SD in (**a–d**, **g**, **i**, **k**). Source data are provided as a Source Data file.

combination of GTPP, in keeping with the notion that oxidative phosphorylation operates as a pro-survival pathway in the context of AURKA inhibition and that this effect may be reversed by inhibitors of the electron transport chain (Fig. 3g). Consistently, we observed a synergistic reduction in proliferation of tumor cells following the combination treatment with alisertib with either GTPP, oligomycin, or metformin (Fig. 3h–k and Fig. S7f–h). Next, we determined whether the combination treatment elicited in part its effects by enhanced induction of cell death. To this purpose, we performed annexin V/PI staining and treated several GBM cells with either vehicle, gamitrinib, alisertib, or the combination of both (Figs. S8a, b). Akin to the viability data, the combination treatment resulted in enhanced induction of cell death. In addition, we appreciated a loss of mitochondrial membrane potential and cleavage of initiator and effector caspases as well as cleavage of PARP which is consistent with induction of a cell death with apoptotic features (Figs. S8c, d and S9a). Moreover, we also assessed the protein levels of downstream regulators of apoptosis, which are orchestrated by anti-apoptotic Bcl-2 family members. While B-cell lymphoma 2 (Bcl-2), B-cell lymphoma-extra-large (Bcl-xL), and myeloid cell leukemia 1 (Mcl-1) were reduced following treatment with the combination treatment, Noxa and ATF4 protein levels were increased (Fig. S9b), overall in keeping with a pro-apoptotic state.

**Aurora kinase A inhibition mediated loss of c-Myc facilitates an increase in PGC1α to drive oxidative metabolism.** Based on a transcriptome analyses, we found that following alisertib exposure GBM22 cells activate PPARA pathways (with increases in PGC1α) and oxidative metabolism accompanied by inhibition of c-MYC signaling (Fig. 4a). Next, we confirmed that PGC1α is indeed upregulated as shown in the microarray analysis. To this purpose, we analyzed mRNA levels of PGC1α by real-time PCR analysis following acute and chronic treatment with alisertib in several GBM cell cultures (Fig. 4b, c). In addition, we confirmed that PDK4, a target of PGC1α, is upregulated as well (Fig. 4d). Next, we assessed the expression of PGC1α and c-Myc at the protein level. We noted that both acute and chronic treatments suppressed c-Myc levels, while enhancing the expression of PGC1α, confirming an inverse correlation of the two transcription factors (Fig. 4e, f). Similarly, silencing by shRNA or CRISPR mediated suppression of ARKA also recapitulate the phenotype elicited by the Aurora kinase A inhibitor (Fig. 4g, h). Next, we determined whether the inverse relationship of c-Myc and PGC1α may be appreciated in patients as well. To this end, we interrogated the TCGA database and found that in GBM c-Myc levels are elevated as compared to normal brain tissue (Fig.

S10a–c). In contrast, PGC1α levels are downregulated in GBM as compared to normal brain tissue. In addition, we considered the mRNA expression of c-Myc and PGC1α in individual tumors (GBM TCGA database, as well as the GBM PDX Mayo Clinic database) and found that high levels of c-Myc were correlated with low levels of PGC1A mRNA in a statistical significant manner (Fig. S10b, c). To elucidate a better understanding, how the transcriptional increase of PGC1α is facilitated by Aurora kinase A inhibition, we performed CHIP-seq and ATAC-seq[19] analysis and checked the PGC1α promoter and enhancer region for potential transcription factor binding sites. We noted that the chromatin accessibility as shown by H3K27ac CHIP-seq and ATAC-seq was increased at a potential c-Myc binding region (Fig. 4i). To confirm this observation, we performed chromatin immunoprecipitation of c-Myc and amplified the PGC1α promoter region in GBM cells either treated with vehicle or with alisertib. While we found strong binding of c-Myc to the PGC1α promoter in vehicle-treated cells, reduced binding was detected following exposure to alisertib (Fig. 4i–k and Fig. S11a). Notably, we found an enhanced acetylation of the same region following alisertib inhibitor treatment, suggesting that c-Myc may act as a suppressor of PGC1α (Fig. 4i–k and Fig. S11a). To further validate this hypothesis, we performed epistasis experiments involving, over-expression and silencing of c-Myc in the context of AURKA inhibition. To this end, we transfected c-Myc and T58A mutant c-Myc in GBM22. Following transfection, GBM22 cells were treated with increasing concentrations of alisertib and assessed for mRNA levels of PGC1α. We noted that forced expression of the T58A c-Myc mutant attenuated alisertib-mediated increase in PGC1α mRNA levels (Fig. 4l). These results were confirmed on the protein level as well (Fig. 4m). Next, we wondered whether c-Myc over-expression could reverse the increase of PGC1α in alisertib-resistant GBM22 cells. Indeed, we observed that adenoviral-mediated over-expression of c-Myc suppressed PGC1α levels (Fig. S11b). Consistently, when we silenced c-Myc there is no further increase of PGC1α level upon alisertib treatment (Fig. 4n and Fig. S11c, d). These findings support the notion that c-Myc is involved in the regulation of PGC1α and that c-Myc potentially acts as a suppressor of PGC1α. Next, we determined the functional role of PGC1α in the context of Aurora kinase A inhibition. To this end, we silenced PGC1α in GBM22 and SF188 cells and treated them with increasing concentrations of alisertib. We found that silencing of PGC1α enhanced the efficacy of alisertib to reduce the cellular viability of the indicated GBM cells (Fig. 4o), suggesting that PGC1α acted as a pro-survival factor in the context of Aurora kinase A inhibition. We hypothesized that PGC1α is at least partially responsible for Aurora kinase A inhibition mediated increase in

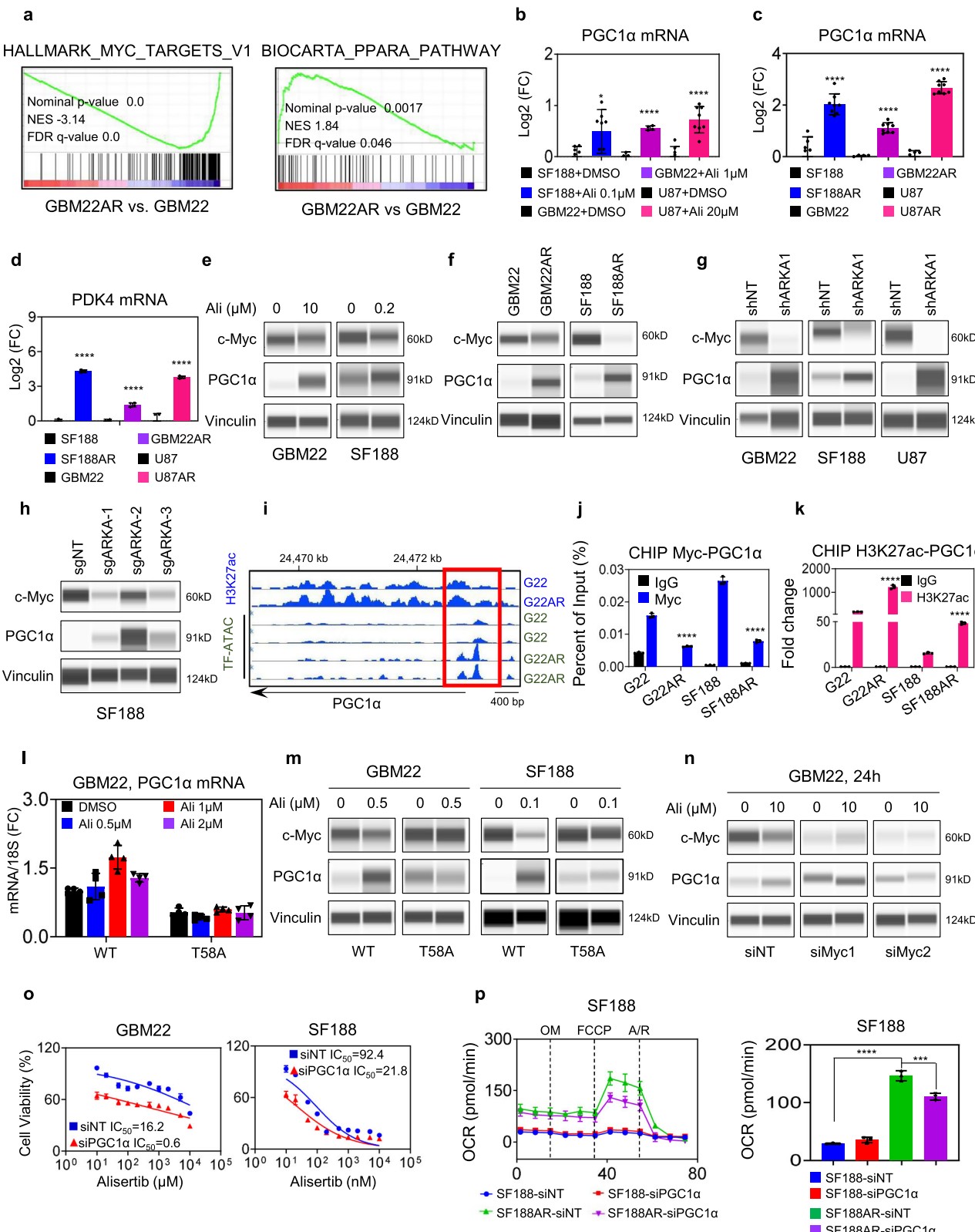

oxygen consumption rate (Fig. 4p). As above, parental or alisertib treated SF188 cells were transfected with non-targeting or PGC1α specific siRNA. Subsequently, cells were subjected to extracellular flux analysis on the seahorse analyzer. While PGC1α had little effect on the oxygen consumption rate in parental SF188 GBM cells, there was a significant reduction in oxidative metabolism in alisertib treated cells, indicating that PGC1α does not appear to play a major role in parental cells, but impacts oxygen consumption in GBM cells exposed to alisertib (Fig. 4p). All in all, these findings support the notion that alisertib-driven activation of oxidative phosphorylation is pro-survival, which in part is mediated by PGC1α. We also validated whether c-Myc silencing would elevate oxygen consumption rate given its suppressive effects on PGC1α and whether loss of c-Myc is responsible in part for the upregulation of the oxygen consumption

**Fig. 4 AURKA inhibition drives PPARA signaling leading to an increase of PGC1α in a partial c-Myc dependent manner. a** Parental or chronically alisertib treated GBM22 cells were subjected to transcriptomic analysis and followed by GSEA. Shown are the enrichment plots of Hallmark_Myc_Target and Biocarta_PPARA_Pathway. NES: normalized enrichment score. FDR: false discovery rate. **b** Real-time PCR analysis of PGC1α mRNA levels in SF188, GBM22, and U87 treated with alisertib or DMSO ($n = 8$ in SF188 and U87, $n = 4$ in GBM22 independent samples) (*$p = 0.0116$, ****$p < 0.0001$). **c** Real-time PCR analysis of PGC1α mRNA levels in parental or alisertib chronically exposed SF188, GBM22, and U87 cells ($n = 8$ independent samples) (****$p < 0.0001$). **d** Real-time PCR analysis of PDK4 mRNA levels of parental or alisertib chronically exposed SF188, GBM22, and U87 cells ($n = 3$ in SF188 and SF188AR, $n = 4$ in GBM22 and GBM22AR, $n = 3$ in U87 and n = 4 in U87AR independent samples) (****$p < 0.0001$). FC: fold change. **e** Protein capillary electrophoresis for the indicated proteins of GBM22 and SF188 cells treated with alisertib for 24 h. Vinculin is used as a loading control. **f** Protein capillary electrophoresis for the indicated proteins of parental or chronically alisertib treated GBM22 or SF188 cells. **g** GBM22, SF188, and U87 cells were transduced with scrambled or shARKA and the whole-cell lysates were subjected to protein capillary electrophoresis with the indicated antibodies. **h** SF188 cells were transduced with scramble or sgAURKA and the whole-cell lysates were subjected for protein capillary electrophoresis with the indicated antibodies. **i** ChIP-sequencing (H3K27ac) and ATAC-sequencing were performed in parental or chronically alisertib treated GBM22 cells. Shown are the respective tracks around the PPARGC1A locus. **j** Parental or chronically alisertib treated GBM22 and SF188 cells were subjected to CHIP with an IgG as a negative control or a c-Myc specific antibody. The PGC1α region was amplified by PCR ($n = 3$ independent samples) (****$p < 0.0001$). **k** Parental or chronically alisertib treated GBM22 and SF188 cells were subjected to CHIP with an IgG as a negative control or a H3K27ac specific antibody. The PGC1α region was amplified ($n = 3$ independent samples) (****$p < 0.0001$). **l** Real-time PCR analysis of PGC1α mRNA levels of GBM22 cells transfected with c-Myc-WT and c-Myc mutant (T58A) followed by treatment with increasing concentrations of alisertib ($n = 4$ independent samples). **m** GBM22 and SF188 cells were transfected with c-Myc-WT and c-Myc mutant (T58A), treated with alisertib for 24 h, and the whole-cell lysates were subjected to protein capillary electrophoresis with the indicated antibodies. **n** GBM22 cells were transfected with non-targeting or two specific siRNAs targeting Myc, treated with 10 μM alisertib for 24 h, and the whole-cell lysates were subjected to protein capillary electrophoresis. **o** GBM22 and SF188 cells were transfected with non-targeting or specific PGC1α siRNA, treated with increasing concentration of alisertib for 72 h, and cellular viability was analyzed ($n = 4$ independent samples). $IC_{50}$ in μM range in GBM22 and in nM range in SF188. **p** Parental or chronically alisertib treated SF188 cells were transfected with non-targeting siRNA or PGC1α specific siRNA and analyzed for oxygen consumption rate (OCR) on a Seahorse XFe24 device. The graph (right panel) shows the OCR level ($n = 3$ independent samples) (***$p = 0.0007$, ****$p < 0.0001$). Statistical significance was assessed by a two-tailed student's $t$-test in (**b–d, j, k**) or ANOVA with Dunnett's multiple comparison test in **p**. Data are shown as mean ± SD in **b–d, j–l, o, p**). Source data are provided as a Source Data file.

rate following silencing of AURKA. To this end, we transfected SF188 GBM cells with siRNA against MYC, AURKA, or the combination (Fig. S11f). Thereafter, we analyzed both oxygen consumption rate and extracellular acidification rate to determine the ratio of the two metabolic indicators. Consistently, we found an elevation of the OCR/ECAR ratio following transfection with all siRNAs, indicating that reduced c-Myc or AURKA levels indeed shift the glycolytic to an oxidative phenotype in GBM cells. The combined silencing did not further enhance this effect, in keeping with the hypothesis that the enhanced oxidative phenotype elicited by silencing AURKA is likely to be a mediated by AURKA driven loss of c-Myc. Given the regulation of both PGC1α and c-Myc on mitochondrial energy metabolism, it was evident to analyze the size of mitochondria following c-Myc and PGC1α modulation in the context of Aurora kinase A inhibition. Silencing of PGC1α suppressed alisertib mediated increase in mitochondrial size as indicated by mito-tracker staining (Fig. S11g). Consistently, adenoviral-mediated over-expression of c-Myc mirrored the findings obtained following genetic interference with PGC1α expression (Fig. S11h), suggesting that c-Myc and PGC1α are implicated in fission and fusion of mitochondria following AURKA inhibition.

**Inhibition of Aurora kinase A reduces both glucose and glutamine oxidation, but leads to enhanced oxidation of palmitic acid.** Localized in the matrix of mitochondria, the citric acid cycle (TCA cycle) is a central metabolic process that has both catabolic and anabolic functions (Fig. 5a). Glucose carbons are metabolized to yield pyruvic acid which either will be converted to oxaloacetate (pyruvate carboxylase) or acetyl-CoA (pyruvate dehydrogenase), while glutamine enters the TCA-cycle via glutamate. Long-chain fatty acids are oxidized in beta-oxidation to yield the final product acetyl-CoA, which condenses with oxaloacetate to give rise to citric acid (Fig. 5a). We wondered how Aurora kinase A inhibition affects metabolic fuel utilization since such information might reveal additional metabolic liabilities. To address this question, we determined the fate of uniformly labeled U-$^{13}$C-glucose, U-$^{13}$C-glutamine, and U-$^{13}$C-palmitic acid carbons following treatment with alisertib in both PDX and established

GBM cells (Fig. 5b–e and Fig. S12a−d). As anticipated and consistent with the extracellular flux analysis, we found that both the m + 3 isotopologue of pyruvate and lactate showed reduced labeling from glucose carbons (Fig. 5c and Fig. S12b). Consistently, the m + 0 isotopologues of pyruvate and lactate were increased, consistent with an overall reduction of labeling of these two metabolites, in keeping with a suppression of the Warburg effect (Fig. 5c and Fig. S12b). While the labeling of the m + 2 citric acid isotopologue was reduced in samples cultured with U-$^{13}$C-glucose we detected an increase in labeling from U-$^{13}$C-palmitic acid. To maintain fatty acid oxidation, anaplerosis is supportive, which is facilitated through pyruvate carboxylase mediated generation of oxaloacetate (from glucose carbons), for which the presence of the m + 3 or m + 5 isotopologue of citric acid serves as a surrogate. While we noted an increase for both isotopologues in U87 GBM cells, the GBM22 PDX cells revealed only a substantial increase in the m + 5 citric acid isotopologue. These findings suggest that enhanced fatty acid oxidation is in part supported through anaplerosis following alisertib inhibition treatment. Another interesting observation is that in U87 GBM cells we found evidence of reductive carboxylation by virtue of an increase of the m + 5 citric acid isotopologue (from glutamine carbons), but this effect was not seen in the GBM22 cells (Fig. 5d and Fig. S12c), indicating that cell-type-specific metabolic fuel utilizations are likely involved. While glutamine fuel utilization was reduced, we found enhanced labeling of amino acids from palmitic acid-derived carbons, in keeping with a key role for fatty acids in Aurora kinase A inhibition mediated metabolic reprogramming in model systems of GBM (Fig. 5e and Fig. S12d).

**Acute and chronic inhibition of Aurora kinase A signaling gives rise to metabolic reliance on fatty acid oxidation.** Our integrated transcriptome, CHIP sequencing, and non-polar metabolites analyses in parental and alisertib treated GBM22 cells demonstrated a profound transcriptional, epigenetic, and metabolic reprogramming of fatty acid oxidation, i.e., upregulation of very long-chain specific acyl-CoA dehydrogenase

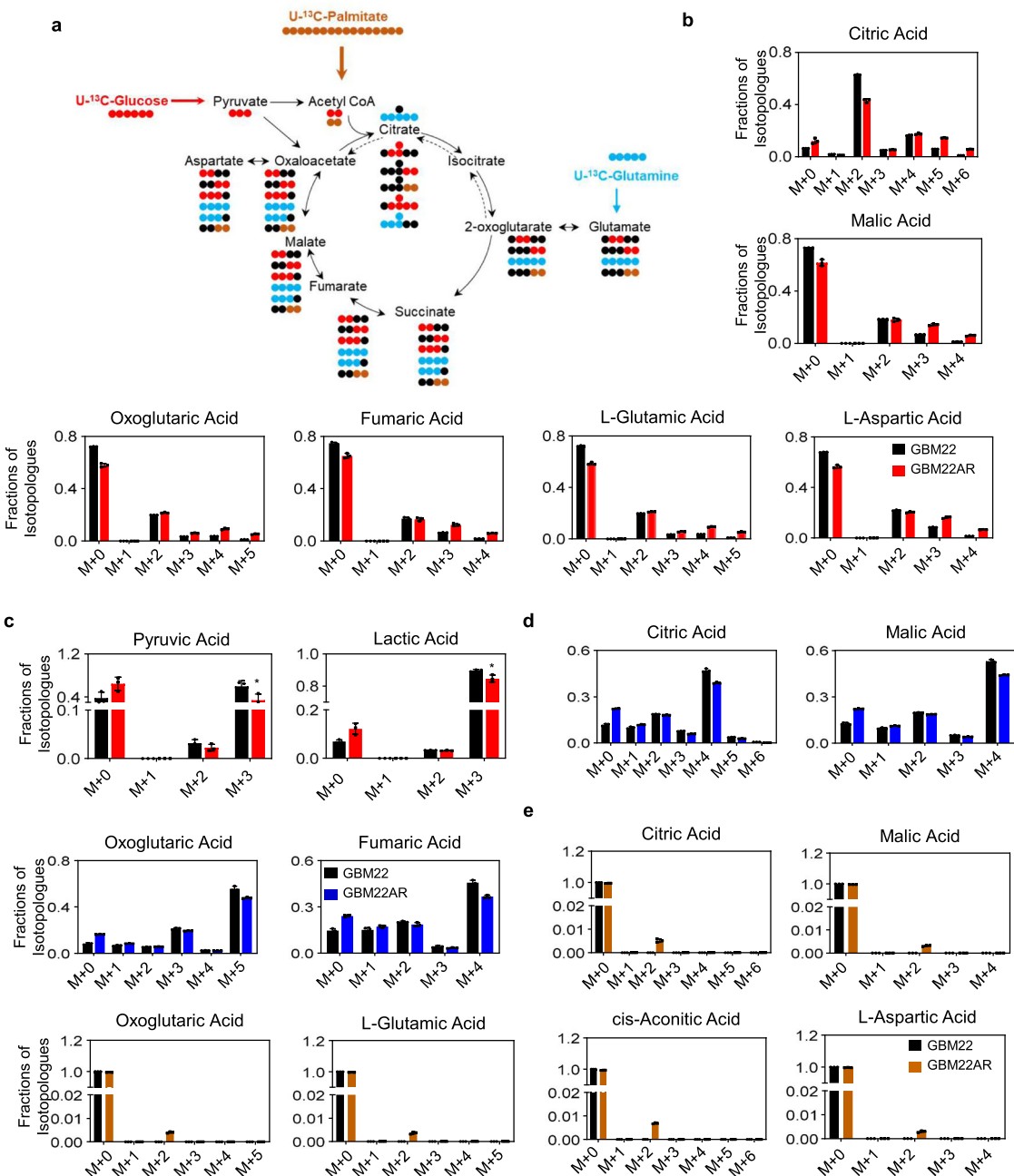

**Fig. 5 AURKA inhibition impacts central carbon metabolism resulting in enhanced labeling of TCA cycle metabolites by long-chain fatty acids. a** Summary of key reactions related to the tracer experiment. Blue circles indicate $^{13}$C carbons from glutamine, brown circles indicate $^{13}$C carbons from palmitic acid, whereas red circles highlight $^{13}$C carbons from glucose. Black circles are used to display $^{12}$C carbons. **b**, **c** Parental or chronically alisertib treated GBM22 cells were cultured in DMEM media containing 25 mM U-$^{13}$C glucose, 4 mM glutamine and 10% dialyzed FBS and were subjected to LC/MS analysis ($n = 3$ independent samples) (pyruvic acid: $*p = 0.0492$, lactic acid: $*p = 0.019$). **d** Parental or chronically alisertib treated GBM22 cells were cultured in DMEM media containing 25 mM glucose, 4 mM U-$^{13}$C glutamine and 10% dialyzed FBS and were subjected to LC/MS analysis ($n = 3$ independent samples). **e** Parental or chronically alisertib treated GBM22 cells were cultured in DMEM media containing 5 mM glucose, 1 mM glutamine, 100 μM U-$^{13}$C palmitic acid, and 10% dialyzed FBS and were subjected to LC/MS analysis. Shown are relative percentages of the isotopologues for each metabolite ($n = 3$ independent samples). Data are shown as mean ± SD in (**b**−**e**). Source data are provided as a Source Data file.

(ACADVL), short/branched-chain specific acyl-CoA dehydrogenase (ACADSB), electron transfer flavoprotein dehydrogenase (ETFDH)) and fatty acids transporters (CPT1C and CD36) (Fig. 6a–f and Fig. S13a−c). The transcriptional findings were confirmed by real-time PCR analysis (Fig. 6c and Fig. S13d). Chip-seq. (H3K27ac) indicated a substantial epigenetic reprogramming with an increase in total peaks following alisertib exposure, which was reflected by enhanced peak formation in

areas close to genes involved in FAO (Fig. S13c). With regards to the non-polar metabolite analyses, we noted a substantial increase of acyl-carnitines and an elevation of lysophosphatidyl derivatives, strongly suggestive of an increase in fatty acid oxidation in alisertib exposed tumor cells and in keeping with our metabolic fuel dependency analysis (Fig. 6d–f). We also assessed whether c-Myc is implicated in the increase of enzymes and transporters related to fatty acid oxidation. To this end, parental and alisertib

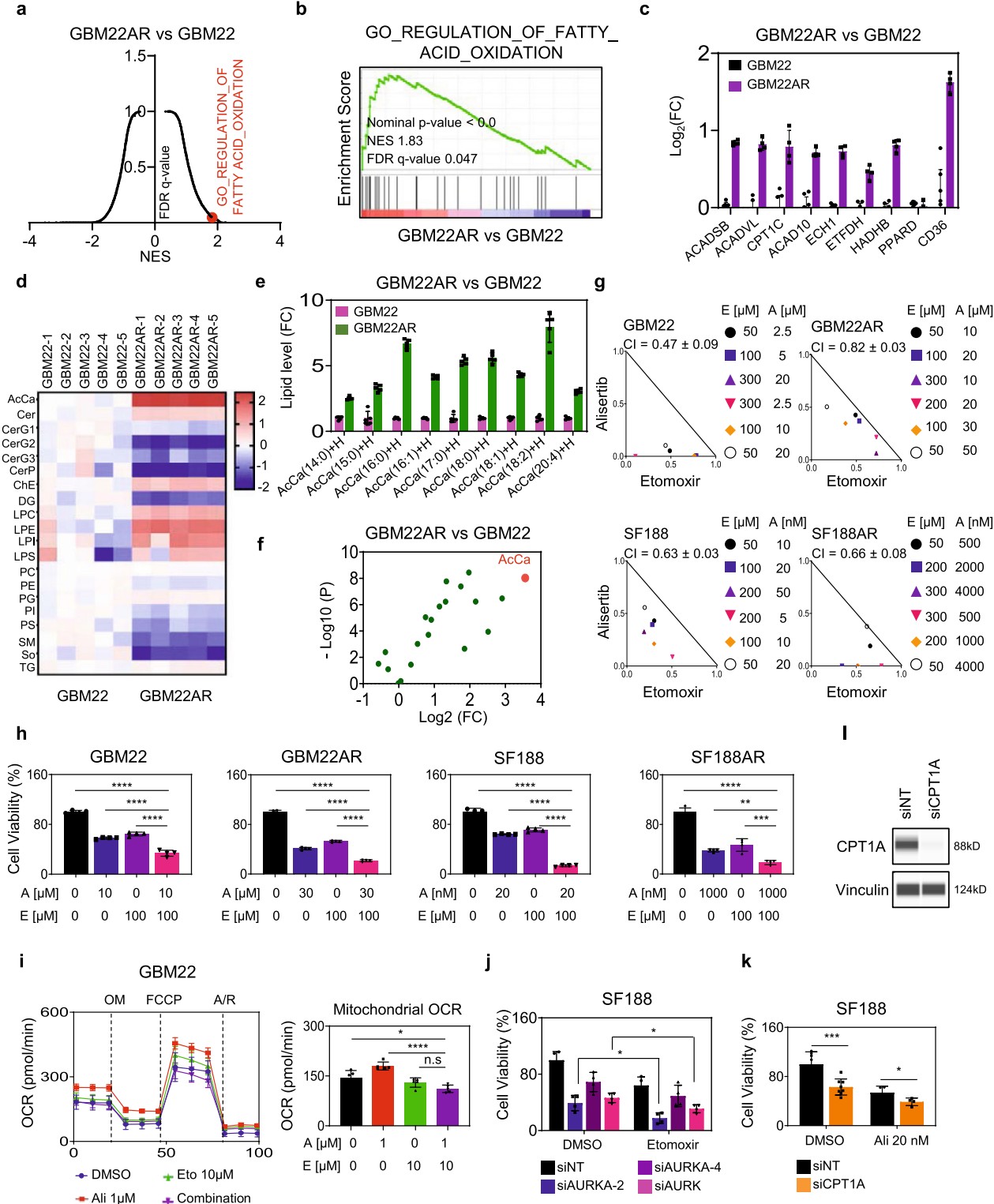

treated GBM22 cells were transduced with a c-Myc adenovirus, and following transduction the mRNA levels of enzymes related to fatty acid oxidation were analyzed. We found that indeed certain enzymes were suppressed following c-Myc over-expression, suggesting that loss of c-Myc mediated by Aurora kinase A inhibition is partially implicated in the engagement of fatty acid oxidation (Fig. S13e). These observations prompted us to test the hypothesis that alisertib along with the clinical validated FAO inhibitor, etomoxir, which binds and blocks the mitochondrial

fatty acid transporter CPT1A, would reduce the cellular viability of GBM cells in a more than additive manner. Indeed, we found that the combination treatment reduced the growth of GBM cells in a synergistic manner (Fig. 6g, h, and Fig. S13f−i). Notably, astrocytes were less responsive to the combination treatment, suggesting a favorable toxicity profile (Fig. S13j). We next asked whether the combination treatment of alisertib and etomoxir resulted in enhanced cell death as well (Fig. S14a−f). Flow cytometry data revealed enhanced apoptosis and dissipation of

**Fig. 6 Dual inhibition of FAO and AURKA elicits a synergistic reduction in cellular viability of GBM cells.** For **a, b** parental or chronically alisertib treated GBM22 cells were subjected to transcriptomic analysis and followed by GSEA. Shown are a graphical plot of NES vs FDR-q value of microarray data in (**a**) and an enrichment plot of Go_Regulation_of_Fatty_Acid_Oxidation in (**b**). NES: normalized enrichment score. FDR: false discovery rate. **c** Real-time PCR analysis of genes related to fatty acid oxidation (mRNA levels) of parental or chronically alisertib treated GBM22 cells ($n = 8$ in GBM22 and $n = 4$ in GBM22AR independent samples). FC: fold change. **d** Non-polar metabolite analysis of parental or chronically alisertib treated GBM22 cells. A heatmap of parental or chronically alisertib treated GBM22 cells is shown. AcCa: acyl carnitine; Cer: ceramides; CerG1-3: neutral glycosphingolipids. CerP: phosphosphingolipids; ChE: cholesterol Ester; DG: diglyceride; LPC: lysophosphatidylcholine; LPE lysophosphatidylethanolamine; LPI: lysophosphatidylinositol; LPS: lysophosphatidylserine; PC: phosphatidylcholine; PE: phosphatidylethanolamine; PG: phosphatidylglycerol; PI: phosphatidylinositol; PS: phosphatidylserine; SM: sphingomyelin; So: sphingosine; TG: triglyceride. For **e**, **f** acyl-carnitine and lysophosphatidyl levels in the non-polar metabolite analysis of parental or chronically alisertib treated GBM22 cells ($n = 5$ independent samples). For **g, h** Parental or chronically alisertib treated GBM22 and SF188 cells were treated with alisertib, etomoxir, or the combination of both for 72 h, and cellular viability was analyzed. Isobolograms are shown in (**g**) and quantification of cell viability is shown in (**h**) ($n = 4$ independent samples) (**p = 0.0047, ***p = 0.0002, ****p < 0.0001$). **i** Parental or chronically alisertib treated GBM22 cells were treated with alisertib, etomoxir, or the combination of both for 24 h and analyzed for oxygen consumption rate (OCR) on a Seahorse XFe24 device. The graph (right panel) shows the OCR level ($n = 5$ independent samples) (*$p = 0.0195$, ****$p < 0.0001$). **j** SF188 cells were transfected with non-targeting siNT or siAURKA (single or pool), treated with etomoxir for 72 h, and cellular viability was analyzed ($n = 4$ independent samples) (siAURKA-2: *$p = 0.0181$, siAURKA-4: *$p = 0.1015$). For **k, l** SF188 cells were transfected with non-targeting siNT or CPT1A specific siRNA (siCPT1A), treated with alisertib for 72 h, and cellular viability was analyzed ($n = 7$ in siNT + DMSO, $n = 8$ in siCPT1A + DMSO, $n = 4$ in siNT + Ali 20 nM and siCPT1A + Ali 20 nM independent samples) (*$p = 0.0495$, ***$p = 0.0009$). Protein capillary electrophoresis confirms the silencing of CPT1A is shown in **l**. Vinculin is used as a loading control. Statistical significance was assessed by a two-tailed student's $t$-test in (**j, k**) or ANOVA with Dunnett's multiple comparison test in (**h, i**). Data are shown as mean ± SD in (**c, e, h−k**). Source data are provided as a Source Data file.

mitochondrial membrane potential in the combination treatment of alisertib and etomoxir as compared to single treatments and vehicle (Fig. S14a−f). Notably, minimal cell death induction was detected by the combination treatment in astrocytes, favoring a tumor-specific effect (Fig. S14c, d). Next, we tested whether the increase in OCR related to Aurora kinase A inhibition is attenuated in the context of the combination treatment with etomoxir. We found that the alisertib-mediated increase in OCR was attenuated in the presence of etomoxir (Fig. 6i). To confirm the specificity of the findings obtained from the drug compounds, we took a two-pronged strategy by silencing their presumed molecular targets, AURKA (alisertib) and CPT1A (etomoxir). Genetic inhibition of AURKA (siRNA) enhanced the efficacy of etomoxir to reduce the cellular viability of GBM cells (Fig. 6j). Conversely, silencing of CPT1A sensitized GBM cells for alisertib-mediated reduction of cellular viability (Fig. 6k, l).

**Dual inhibition of Aurora kinase A along with oxidative energy metabolism extends overall survival in PDX models of glioblastoma in mice.** Considering the in vitro findings related to the various drug combinations, involving Aurora kinase A inhibition and targeting of tumor metabolism we assessed whether these observations bear translational relevance. To this purpose, we harnessed PDX models since they are currently the premier model system to study drug effects in a preclinical setting in vivo. The appeal of PDX tumor model systems is based on the fundamental principle that they keep most features of the primary tumors and therefore likely provide the closest resemblance to a patient scenario. First, we initiated the in vivo studies with two subcutaneous GBM patient-derived xenograft models, GBM12 and GBM43. Following the establishment of tumors, mice were randomly divided into four groups: vehicle, alisertib (30 mg/kg), etomoxir (20 mg/kg), and the combination of both. We detected a stronger suppression in tumor growth in animals that received both alisertib and etomoxir treatment as compared to the single treatments in both the GBM12 and GBM43 models (Fig. 7a−d).

In addition, we determined whether a combined inhibition of mitochondrial Hsp90 and Aurora kinase A leads to enhanced growth inhibition in PDX models of GBM. To this end, we tested the combination treatment of gamitrinib and alisertib in the GBM12 PDX model. Akin to the findings related to the FAO inhibitor, the combination treatment of alisertib and gamitrinib reduced tumor growth more potently than single treatment with

alisertib (Fig. S15a, b). Having observed substantial tumor-suppressive effects in model systems of malignant glioma, we sought to determine whether these combination treatments are similarly efficacious in other solid malignant tumors as well. To this end, we chose colonic adenocarcinoma as a common solid malignancy and utilized the HCT116 xenograft model to assess the various treatments in vivo. Following the establishment of tumors, six groups were formed, consisting of vehicle, alisertib (30 mg/kg), etomoxir (20 mg/kg), gamitrinib (5 mg/kg), alisertib + gamitrinib, and alisertib + etomoxir. Similar to the GBM models the combination treatments reduced the growth of the tumor cells more potently than the single treatments, suggesting that the approach to combine metabolic inhibitors along with Aurora kinase A blockers are likely applicable to a broader range of solid malignancies (Fig. S15c, d).

Next, we assessed whether the metabolic in vitro findings are observed in the animals as well. To this purpose, we injected tumor (GBM12 PDX) bearing animals from both the vehicle and alisertib treated group with U-13C-palmitic acid and submitted the tumors for LC/MS analysis. We found that tumors treated with alisertib displayed enhanced labeling of both TCA-cycle metabolites, as well as amino acids from palmitic acid carbons, in keeping with the notion that Aurora kinase A inhibition facilitates fatty acid oxidation in vivo as well (Fig. 7e). In addition, we assessed whether in vivo PGC1α is upregulated following treatment with alisertib. We found that PGC1α was increased and more abundantly present in tumors treated with alisertib (Fig. 7f), indicating that likely PGC1α is implicated in the metabolic reprogramming in vivo as well.

To gain more insight about the impact of the drug treatments on the cellular density, proliferation, and cell death (necrosis and apoptosis), representative tumors were paraffin-embedded and stained with hematoxylin and eosin (H&E). While vehicle, etomoxir, and alisertib-treated tumors displayed a dense cellular architecture with brisk mitotic activity, the combination treatment showed lower cellular density with a reduction of mitosis and areas of cell death (Fig. S15e). To confirm cell death induction by the combination treatment, TUNEL staining was performed (Fig. S15f−h). We detected more TUNEL positive cells in tumors exposed to the combination treatment as compared to vehicle or single-drug treatments. Consistently, the proliferation index (Ki67) was noted lesser in the combination treatment (Fig. S15f−h).

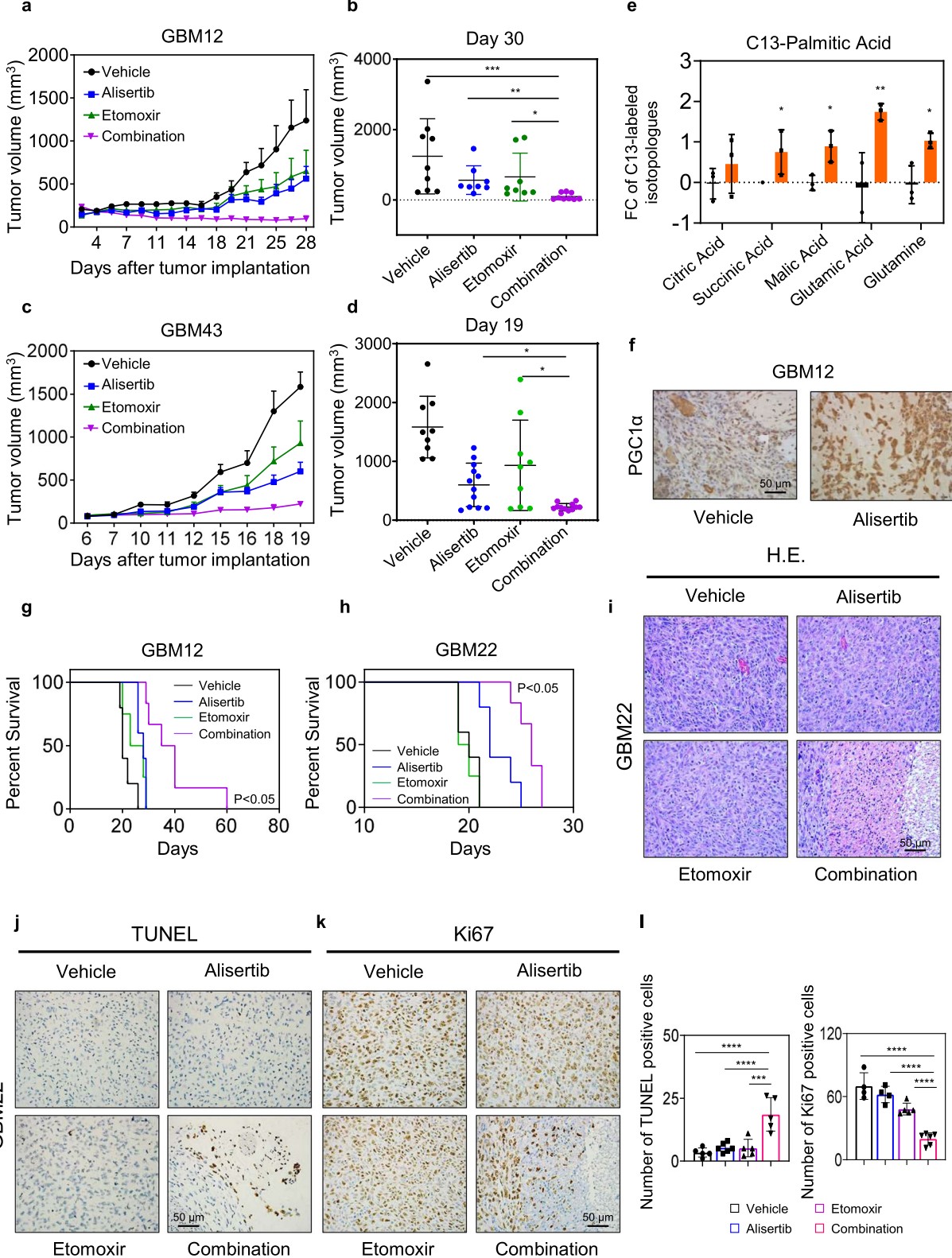

These findings are in keeping with the reduced size of the tumors in the combination treatment. Similar findings were made in tumors that were treated with vehicle, gamitrinib, alisertib, or the combination of both (Fig. S15i−l). The integrity of solid organs such as heart or kidney was assessed and we did not observe induction of detectable toxicity in the combination treatment (Fig. S16a).

Given the significant promise in the heterotopic GBM xenograft studies, we extended our study to orthotopic PDX models of glioblastoma given that these model systems, though with some limitations, are currently considered to be in closest resemblance to the patient scenario. To this purpose, we utilized the GBM12 and GBM22 PDX orthotopic xenograft models, respectively. After the establishment of tumors, four groups of

**Fig. 7 Dual inhibition of FAO and AURKA extends animal survival in orthotopic patient-derived xenograft models of human GBM.** For **a, b** GBM12 cells were implanted into the subcutis of immunocompromised Nu/Nu mice. After the tumors were established, the mice were randomized to four treatment groups: vehicle, alisertib (30 mg/kg), etomoxir (20 mg/kg), and the combination treatment. The tumor volume over time is shown in (**a**) and the tumor volume on the last day of the experiment is shown in (**b**) ($n = 8$ in alisertib and etomoxir, $n = 9$ in vehicle and combination independent tumors) (*$p = 0.0438$, **$p = 0.0085$, ***$p = 0.0003$). For **c, d** GBM43 cells were implanted into the subcutis of immunocompromised Nu/Nu mice. After the tumors were established, the mice were randomized to four treatment groups: vehicle, alisertib (30 mg/kg), etomoxir (20 mg/kg), and combination treatment of both. The tumor volume over time is shown in (**c**) and the tumor volume on the last day of the experiment is shown in (**d**) ($n = 9$ in vehicle and etomoxir, $n = 12$ in alisertib and combination independent tumors) (Ali vs Combination: *$p = 0.0462$, etomoxir vs combination: *$p = 0.0162$). **e** GBM12 cells were implanted and treated with alisertib, etomoxir, or the combination of both as described in (**a**). On the last day of the experiment, mice were injected with 100 μM U-$^{13}$C-palmitic acid for 4 h and tumors were subjected to LC/MS. The graph shows the U-$^{13}$C-palmitic acid labeling of the TCA cycle. FC: fold change. **f** Tumors from the experiment in (**a**) were fixed and stained with PGC1α. Scale bar: 50 μM. For **g, h** GBM12 and GBM22 cells were implanted in the right striatum of nude mice. Four groups were randomly assigned: vehicle, alisertib, etomoxir, and combination of both, seven days after the implantation. Mice were treated three times per week and animal survival is provided (Kaplan−Meier-curve): vehicle: GBM12: 20d, GBM22: 20d; alisertib: GBM12: 28d, GBM22: 22d; etomoxir: GBM12: 25.5d, GBM22: 19.5d; combination: GBM12: 37.5d, GBM22: 26d. The log-rank test was used to assess statistical significance ($n = 5$ in vehicle and alisertib, $n = 4$ in etomoxir, $n = 6$ in combination). For (**i**−**k**) The brain tumors from the experiment in (**h**) were fixed and stained with H&E, TUNEL, or Ki67. Scale bar: 50 μm. **l** Quantification of TUNEL and Ki67 positive cells in (**j, k**), respectively ($n = 5$ in vehicle, etomoxir, combination, $n = 6$ in alisertib independent high-power field microscopy) (***$p = 0.0001$, ****$p < 0.0001$). Statistical significance was assessed by a two-tailed student's $t$-test in **e** or ANOVA with Dunnett's multiple comparison test in (**b, d, l**). Data are shown as mean ± SEM in (**a, c**) and as mean ± SD in (**b, d, e, l**). Source data are provided as a Source Data file.

animals were formed: vehicle, alisertib, etomoxir, and combination of both (Fig. 7g, h). In both models, we found that the combination treatment extended animal survival significantly longer than compared to a single treatment with alisertib (Fig. 7g, h), suggesting potential clinical efficacy. Consistently, orthotopic tumors that received treatment with the combination treatment were smaller than vehicle or single-agent treated tumors as indicated by histopathological evaluation (Fig. 7i–l and Fig. S16b). In addition, they displayed an increase in TUNEL positive GBM cells and a reduced amount of Ki67 positive GBM cells as compared to single treatments (Fig. 7l). Collectively, our data provide a foundation for further evaluation and potential clinical consideration of drug combination therapies, involving AURKA and FAO/OXPHOS inhibitors.

## Discussion
The exploration of tumor cell metabolism has recently gained more traction in cancer research[1,5,8,18,20−35]. Through the recent advancement in high throughput, genetic techniques and medicinal chemistry direct targeting of key enzymes of metabolism have become more achievable and tractable than in the past. However, brain tumor metabolism has been relatively understudied and it appears likely that additional knowledge in this field will foster treatment approaches for these devastating diseases. The general dogma of cancer cell metabolism is centered on aerobic glycolysis (Warburg-effect), but how this effect in detail is regulated remains enigmatic. The resolution of this complex interplay will likely yield substantial insights into the pathophysiology of solid and non-solid malignancies and the design of more efficient therapies. In this study, we made the unique observation that Aurora kinase A inhibition is synthetically lethal with the interference of the respiratory transport chain in solid malignancies. While Aurora kinase A is considered to modulate cell proliferation through modulation of mitosis, we here made the unexpected, but the important discovery that Aurora kinase A inhibition stalled the proliferation of GBM cells in a manner that was highly dependent on glycolysis. Whereas Aurora kinase A inhibition resulted in suppression of glycolysis, it concomitantly activated oxidative metabolism, suggesting that metabolic reprogramming might represent an escape mechanism of GBM cells from therapy. When cells were cultured in galactose, they were significantly more resistant towards reduction of cellular viability mediated by Aurora kinase A inhibitors, suggesting that tumor cells with an oxidative energy metabolism will display resistance

towards AURKA inhibitor treatment. Consistently, alisertib-resistant GBM cells displayed a hyper oxidative phenotype. These findings are significant because it is known that tumor cells may display a range of dependency on either glycolysis or oxidative phosphorylation, which nowadays may be assessed rapidly. Therefore, the grouping of tumors according to their metabolism might predict response and resistance to therapies, involving AURKA inhibitors. These results are in contrast with other inhibitors that target oxidative energy metabolism, e.g., LXR agonists and CLPP protease activators[36−38]. In these instances, galactose exposure renders tumor cells more susceptible to such compounds, reinforcing the role of this approach to distinguish between glycolytic or oxidative energy metabolism dependency. Given that there is a broad array of compounds available that block oxidative metabolism, we utilized classical inhibitors of oxidative metabolism to demonstrate that dual inhibition of the electron transport chain and Aurora kinase A inhibition is synthetically lethal in a model system of GBM and in a model system of colorectal carcinoma. To the best of our knowledge, these findings have not been reported elsewhere before even not in other tumor entities. Given that we have validated our results beyond GBM models (i.e., in a colonic carcinoma xenograft) there is a reasonable likelihood that other solid tumors may respond in a similar fashion.

Aerobic glycolysis is controlled by c-Myc and interference with this transcription factor has been shown to suppress glycolysis and in turn tumor growth in a variety of different tumor model system even in contexts where MYC is not amplified[9]. Our proteomic analysis suggested that Aurora kinase A inhibitors blocked c-Myc levels. Silencing and over-expression of c-Myc implied a causal role for c-Myc. Ablating c-Myc from GBM cells resulted in enhanced resistance towards Aurora kinase A inhibition mediated reduction in cellular viability. Consistently, modest over-expression of c-Myc rescued partially rescued from alisertib mediated loss in cellular viability, consistent with earlier findings in the context of neuroblastoma (MYCN/AURKA)[39]. Co-immunoprecipitation studies further suggested a direct interaction between Aurora kinase A and c-Myc and that Aurora kinase A counteracted c-Myc degradation by the proteasome. Our results provide evidence of this interaction in these settings. Moreover, both the fact that AURKA signaling regulates tumor cell metabolism through c-Myc is important and that the response to AURKA inhibitors relies on tumor metabolism (balance between glycolysis and oxidative phosphorylation) via

c-Myc is relevant even beyond the current studies given that any treatment modality, involving AURKA inhibitors, will be most likely influenced by the metabolic status of the tumor. Prior results in other tumor entities pointed towards a role of Aurora kinase A in regulating glycolysis and oxidative energy metabolism, but our results are unique due to the identified signaling pathway and its potential translational implications[40–42].

While glycolysis and c-Myc levels declined following Aurora kinase A pathway perturbation, we noted an increase of the oxygen consumption rate accompanied by an elevation of PGC1α, suggesting both an inverse correlation between glycolysis and oxidative metabolism, as well as between c-Myc and PGC1α. Following ATAC-seq. analysis of the PGC1α promoter region we noted enhanced accessibility at a MYC binding region. Our experiments strongly suggest that c-MYC acted as a repressor of PGC1α expression in a PDX model system of GBM. Importantly, PGC1α caused resistance to Aurora kinase A inhibition mediated reduction in cellular viability, which up to date has not been described before. Given the existence of PGC1α inhibitors and considering our findings it may also be tempting to speculate whether such drugs could synergize with AURKA inhibitors to reduce the proliferation of tumor cells.

The following interference with AURKA signaling our comprehensive metabolite analysis revealed strong evidence of reliance of GBM cells on fatty acid oxidation, shifting away from their glycolytic phenotype. Indeed, extracellular flux and carbon tracing analyses confirmed this notion, and inhibition of FAO or the electron transport chain along with Aurora kinase A blockage is synthetically lethal in GBM model systems and other recalcitrant solid malignancies, e.g., colon carcinoma. Utilizing the current gold standard for preclinical drug assessment for GBM, we were able to demonstrate that the combination treatment of alisertib with inhibitors of FAO extended animal survival. All in all, our findings reinforce the concept that targeting drug-induced reprogrammed GBM metabolism is feasible, which may have future therapeutic implications.

## Methods

**Cell cultures and growth conditions**. All cell lines were incubated at 37 °C and were maintained in an atmosphere containing 5% $CO_2$. GBM22, GBM12, and GBM43 cells were obtained from Dr. Jann Sarkaria (Mayo Clinic, Rochester, MN) between 2014 and 2020. GBM12 is a PDX line, which is derived from a male patient (glioblastoma, IDH-wildtype, mesenchymal molecular subtype, EGFR amplified, PTEN wt, TP53 mutated (splice mutation), MYC non-amplified). GBM22 is a PDX line, which is derived from a male patient (Gliosarcoma, IDH-wildtype, classical molecular subtype, IDH1 wt, EGFR gain, PTEN wt, TP53 mutated, MYC amplified). GBM43 is a PDX line, which is derived from a male patient (Glioblastoma, IDH-wildtype, classical molecular subtype, IDH wt, EGFR gain, PTEN wildtype, TP53 mutated, MYC non-amplified). U87 and HCT116 cell lines were obtained from the American Type Culture Collection (Manassas, VA). Cells were cultured in DMEM (Fisher Scientific, MT10013CV), 10% FBS (Gemini) and 100 µg/ml of Primocin (Invivogen, ant-pm-1). SF188 cells were obtained from UCSF (CA) and were cultured in DMEM, 10% FBS, 1% Glutamax (Thermo Fisher, 35050061), and 100 µg/ml of primocin. SF188 is a pediatric glioblastoma cell line derived from 8-year old male (TP53 mut, PTEN wt, MYC-amplified). Astrocyte cells were obtained from ScienceCell Research Laboratories (Carlsbad, CA) and were cultured in DMEM, 10% FBS, primocin, and N2 supplement (Thermo Fisher, 17502048). Alisertib resistant cells were established by continuous exposure to alisertib for 10 days. For the treatment experiment, cells were cultured in DMEM containing 1.5% FBS. The respective cell line depository authenticated the cells.

**Reagents**. Alisertib (MLN8237), etomoxir (S8244), metformin (S1950), and oligomycin (S1478) were purchased from Selleckchem (Houston, TX). Gamitrinib (GTPP) was provided by Dr. Altieri (Wistar Institute, Philadelphia, PA). Galactose was purchased from Sigma (G0750, Burlington, MA). A 10 mM working solution in dimethylsulfoxide (DMSO) was prepared for all reagents prior to storage at −20 °C. Final concentrations of DMSO were below 0.1% (v/v). MG-132 (474791-5MG) and Cycloheximide (239764-100MG) were purchased from Sigma.

**Plasmid constructs**. AURKA-Untagged (EX-Q0008-M02) and HA-AURKA (EX-Q0008-M06) were purchased from GeneCopoeia (Rockville, MD). pCDH-MSCV-

T2A-puro, pCDH-MSCV-T2A-puro-Flag-Myc WT, and pCDH-MSCV-T2A-puro-Flag-Myc T58A were generously given by Dr. Shideng Bao (Cleveland Clinic, Cleveland, OH), pcDNA6-N-3XFLAG-GSK3β (Cat #123592), FLAG-HA-pcDNA3.1 (Cat# 52535), and pCDNA3-HA-HA-humanCMYC (Cat #74164) were purchased from Addgene (Watertown, Massachusetts).

**Cell viability assays**. Cells were seeded in a 96-well plate with density $3 \times 10^3$ cells per well and allowed to attach overnight. To examine cellular proliferation, CellTiter-Glo® assays were performed according to the manufacturer's instructions (Promega, G7571) and the signal was read by using the SpectraMax i3x (Molecular Device, San Jose, CA). To determine drug synergism the median effect equation (Chou—Talalay) was employed (isobolograms and the combination index (CI)).

**Flow cytometry**. Cells were seeded in a 12 well plate with density $3 \times 10^4$ cells per well and allowed to attach overnight. To detect the apoptosis and necrosis, cells were incubated with FITC Annexin V/ propidium iodide according to the manufacturer's instructions (BD Biosciences, 556420). To detect intrinsic apoptosis staining and loss of mitochondrial membrane potential, TMRE (tetramethylrhodamine ethyl ester perchlorate) staining was performed according to the manufacturer's instructions (CST, 13296S). Signal was detected on a LSRII flow cytometry (Becton–Dickinson, New Jersey) and the data were analyzed with FlowJo software version 10.7 (Tree Star, Ashland, OR). The cells were gated through forward scatter (FSC) and side scatter (SSC) and then analyzed for Annexin V and PI fluorescence. Alternatively, cells were analyzed for TMRE fluorescence.

**Western blot and protein capillary electrophoresis**. For co-IP experiments, cells were lysed in a lysis buffer, containing 100 mM NaCl, 50 mM Tris-HCl, 0.5% NP-40, and 1× protease and phosphatase inhibitor cocktail (Thermo Fisher, 78440). Cell lysates were incubated with IgG or specific antibodies conjugated to Dynabeads™ Protein G for Immunoprecipitation (Thermo Fisher, 10004D) over night at 4 °C. For standard western blot and capillary electrophoresis, cells were lysed directly in the laemmli buffer (Biorad) containing 1× protease and phosphatase inhibitor cocktail (Thermo Fisher, 78440). Samples were run on a 4−12% SDS PAGE gel (Invitrogen, NP0321BOX), the proteins were transferred to a PVDF membrane, and the membrane was blocked with 5% BSA in TBST (0.1% Tween20) and probed with target antibodies. The western blots were acquired by using the Azure (C300) imaging system (Azure Biosystems). Capillary electrophoresis was performed on the Wes instrument according to the manufacturer's instructions (ProteinSimple, San Jose, CA; SM-W004). For standard western blot the primary antibodies used: rabbit anti-PARP (Cell Signaling Technology (CST) 9532; 1:500); rabbit anti-cCP9 (CST 7237; 1:500); rabbit anti-cCP3 (CST 9665; 1:500); rabbit anti-USP9x (CST 5751; 1:500); rabbit anti-Bcl-xL (CST 2764; 1:500); rabbit anti-Bcl-2 (CST 4223; 1:500); rabbit anti-Mcl-1 (CST 5453; 1:500); rabbit anti-CPT1A (CST 12252; 1:500); mouse anti-Noxa (Calbiochem OP180, clone 114C307; 1:500); rabbit anti-BIM (CST 2933; 1:500); mouse anti-β-actin (Sigma Aldrich A1978, clone AC15; 1:8,000); rabbit anti-Aurora A/AIK (1G4) (CST 4718; 1:500); rabbit anti-p-Aurora A (Thr288) (CST 3079; 1:500); rabbit anti-Myc (CST 13987; 1:500); and rabbit anti-GSK-3β (27C10) (CST 9315; 1:500). The secondary antibodies were used: anti-rabbit IgG (H + L) secondary antibody, HRP (Thermo Fisher 31460; 1: 3000) and anti-mouse IgG (H + L) secondary antibody, HRP(Thermo Fisher 31460; 1: 3000). For protein capillary electrophoresis the primary antibodies used rabbit anti-HK2 (CST 2106, 1:25); rabbit anti-Glut1 (CST 12939; 1:25); rabbit anti-LDHA (CST 3582, 1:400); rabbit anti-PGC1α (Novus Biologicals NBP1-04676, 1:25); mouse anti-c-Myc (CST 13987, 1:25); mouse anti-Bcl-2 (R&D System MAB827, 1:25); rabbit anti-ATF4 (CST 11815; 1:25); rabbit anti-Vinculin (Abcam ab129002, 1:500); rabbit anti-Aurora A/AIK (1G4) (CST 4718; 1:500); rabbit anti-GSK-3β (27C10) (CST 9315; 1:25), and rabbit anti-phospho-GSK-3β (Ser9) (CST 5558S; 1:25). The secondary antibodies were used with the manufacturer's instructions: Anti-Rabbit Secondary HRP Antibody (ProteinSimple 042-206) and Anti-Mouse Secondary HRP Antibody (ProteinSimple 042-205).

**Site directed mutagenesis**. HA-Aurora-D274N constructs was generated by using QuikChange II XL Site-Directed Mutagenesis Kit (Cat# 200521) (Agilent, Santa Clara, CA) in accordance with the accompanied instructions manual. Primer sequences are in the Supplementary Table 1.

**Real-time PCR analysis**. Total RNA was isolated using miRNAeasy Mini Kit (QIAGEN 217004) and reverse-transcribed to cDNA using cDNA synthesis kit (Quantabio 101414-106). The cDNA was amplified using power SYBR green RT-PCR reagents kit (Quantabio 101414-276). The reaction was run at 95 °C for 10 min, followed by 40 cycles of 95 °C for 15 s, 60 °C for 30 s, and 72 °C for 30 s, on a qPCR Instrument (Quantabio). All RT-PCR was performed in quadruplicate and the average fold changes were calculated based on 18S in the threshold cycle (Cq). Primer sequences are in the Supplementary Table 1.

**Liquid chromatography and mass spectrometry (LC/MS) and isotope tracing**. LC/MS was performed as described in refs. [43,44]. Briefly, cells were seeded in six-well plates and were extracted by methanol/water/chloroform (600 µL/300 µL/400 µL),

containing an internal standard (Metabolomics Amino Acid Mix Standard, MSK-A2-1.2, Cambridge Isotope Laboratories) for polar and non-polar metabolite extraction. Two layers were separated by drying under nitrogen, the lipid fraction (non-polar) was dissolved in a mixture of acetonitrile, 2-propanol, and water (65:30:5, v/v/v), and the water was used for the solubilization of the polar fraction. Liquid chromatography–high-resolution mass spectrometry (LC/HRMS) was utilized for the analysis of the two fractions. With regards to the isotope tracing experiments, the GBM cells were exposed to DMEM devoid of nutrients (Thermo Fisher, A1443001) for 1 h. After that, the GBM cells were cultured in DMEM, containing either 25 mM (U-$^{13}$C6) D-Glucose (Cambridge Isotope Laboratories, Inc), 4 mM (U-$^{13}$C5) L-Glutamine (Cambridge Isotope Laboratories, Inc), or 100 μM (U-$^{13}$C16) palmitic acid (Cambridge Isotope Laboratories, Inc) for 24 h in the presence of 1.5% dialyzed FBS (Thermo Fisher, A3382001). Polar metabolites were analyzed by Metabolomics Core Facility at Weil Cornell (New York).

**Chromatin immunoprecipitation (CHIP) RT-PCR, CHIP-sequencing, and ATAC-sequencing**. Enzymatic CHIP assays were performed in accordance with the accompanied instructions manual (SimpleChIP® Enzymatic Chromatin IP Kit, CST 9003). Chromatin immunoprecipitation were incubated with H3K27ac antibody (CST 4535, 10 μL/sample), Myc antibody (CST 13987, 10 μL/sample), or Rabbit IgG antibody (CST 2729, 1 μL/sample). CHIP H3K27ac or Myc were submitted for library preparation followed by next-generation sequencing (Illumina HiSeQ 4000 instrument; single read 50 bp (SR50)) (Genewiz, New Jersey). For ATAC-sequencing, cells were extracted and library preparation was performed according to the instructions provided in the ATAC-Seq Kit (Active Motif 53150). Sequencing data were analyzed by Galaxy software (https://usegalaxy.org/) or basepair software (https://www.basepairtech.com/) and were mapped to the human genome (hg38), using bowtie. The raw sequences and bigwig files for ATAC-sequencing and CHIP-sequencing are deposited in GEO: GSE161572 and GSE161573, respectively. Primer sequences are in the Supplementary Table 1.

**Electron microscopy**. For fixation, glioblastoma cells were incubated with 2.5% glutaraldehyde in Cacodylate buffer for 60 min at RT. One percent of osmium tetroxide was used for extended fixation in the same buffer. For the embedding process ALx-112 (Ladd Research Industries, Inc.) and Embed-812 (EMS, Fort Washington, PA) were used. Tissue sections were derived from the MT-Power-Trome XL ultramicrotome (60 nm) followed by staining with Uranyl acetate and lead citrate. Examination of the section was performed on the JEOL JEM-1200 EXII electron microscope. Recorded images were derived from the ORCA-HR digital camera (Hamamatsu).

**Transfections**. Cells were seeded in a 12 well plate with density $5 \times 10^4$ cells per well one day prior to transfection. For siRNA transfection experiments, cells were performed with Lipofectamine RNAiMAX (Invitrogen, 13778075) according to manufacturer's instructions. Non-targeting siRNA pool (D-001810-10-20); AURKA pool siRNA (L-003545-01-0005); AURKA-2 siRNA (J-003545-27-0005); AURKA-4 siRNA (J-003545-29-0005); PGC1α siRNA (L-005111-00-0005), and CPT1A siRNA (L-009749-00-0005) were purchased from Dharmacon. Myc-1 siRNA (CST 6341) and Myc-2 siRNA (CST 6552) were purchased from Cell Signaling. For ectopically expressed experiments, cells were transfected using Lipofectamine 3000 (Invitrogen). pD40-His/V5-c-Myc (#45597) and pD40-His/V5-c-MycT58A (#45598) constructs were purchased from Addgene (Watertown, MA). For knock down and knock out experiments, cells were infected in the presence of 8 μg/mL polybrene (Santa Cruz Biotechnology) and were selected with puromycin (Santa Cruz Biotechnology). ARK-1 shRNA (h) Lentiviral Particles (sc-29731-V) was purchased were purchased from Santa Cruz Biotech (Santa Cruz, CA). AURKA sgRNA CRISPR Lentivirus set (human) (Cat# 128661110101) was purchased from Applied Biological Materials. Adeno CMV Null Adenovirus (Ad-CMV-Null 1300) and c-Myc Adenovirus (Ad-c-Myc 1285) were purchased from Vector Biolabs (Malvern, PA) and were infected according to the manufactures' instructions.

**Microarray and subsequent gene set enrichment (GSEA) analysis**. The experiment is comprised of four Human Gene 2.0 ST arrays, comparing between the control and treated cells. For normalization of the microarrays (performed at the same time), the Robust Multiarray Average (RMA) algorithm along with a CDF (Chip Definition File), which provides the unique Entrez Gene identifiers, were used. The experiments used in this study for GBM22 and GBM22AR were deposited at GEO: GSE152612.

**Extracellular flux analysis**. Oxygen consumption rate (OCR) and extracellular acidification rate (ECAR) were measured with a Seahorse XFe24 Analyzer (Agilent, Santa Clara, CA) or a Seahorse XFp Analyzer (Agilent, CA). Cells were seeded in XFe24 cell culture microplates (Agilent, CA) at $3 \times 10^4$ cells/well or in XFp cell culture microplates at $5 \times 10^3$ cells/well in DMEM medium containing 5 mM glucose, 1 mM pyruvate, and 10% FBS and allowed to attach overnight. Cells were treated with reagents or corresponding solvents in the medium containing 5 mM

glucose, 1 mM pyruvate, and 1.5% FBS in the following day. Mitochondrial stress assay (Agilent, 103015-100) was performed in the Seahorse XF base medium (Agilent, 102353-100) containing 10 mM glucose, 2 mM glutamine, 1 mM pyruvate. The following compounds were injected in a sequential order: 2 μM oligomycin (OM), 2 μM Carbonyl cyanide-4 (trifluoromethoxy) phenylhydrazone (FCCP), and 0.5 μM rotenone/antimycin (R/A). Glycolysis stress assay (Agilent, 103020-100) was performed in the Seahorse XF base medium containing 1 mM glutamine. The following compounds were injected in a sequential order: 10 mM glucose (G), 1 μM oligomycin (O), and 50 mM 2-DG.

**Subcutaneous xenograft model**. $1 \times 10^6$ patient-derived xenograft GBM12, GBM43 cells, or colon carcinoma HCT116 cells were implanted subcutaneously into the flanks of 6−8 weeks old SCID/SHO mice. Intraperitoneal treatments and tumor measurements were performed three times a week. Drugs were dissolved in a mixture of drug, Kolliphor® EL (Sigma, C5135-500G), Ethyl Alcohol (Pharmco-Aaper, 200 Proof), and PBS at the ratio: 10:32:8:50 (v/v/v). Drug doses used: 30 mg/kg alisertib, 20 mg/kg etomoxir or 5 mg/kg GTPP. Tumor size was measured with a caliper and was calculated as (length × width$^2$)/2. Mice body weights were monitored at each time point.

**Orthotopic glioblastoma PDX model**. GBM12 and GBM22 were intracranially injected $30 \times 10^4$ cells at (3 mm lateral, 1 mm anterior of the bregma, and 3 mm down). Intraperitoneal treatments were performed three times a week until the animals become moribund or neurological deficits (retardation, paresis, lethargy, seizures, hyperactivity). To calculate $p$-values (significant $p < 0.05$), the Log-rank test was used. Survival is indicated by Kaplan−Meier survival fractions (plot survival versus time).

**In situ proximity ligation assay (PLA)**. Cells were seeded with density 1,500 cells/well on the eight well chamber Nunc® Lab-Tek® Chamber Slide™ system (Sigma, C7182-1PAK) overnight. Cells were fixed in 4% formaldehyde (Thermo Fisher, 28908) for 10 min at RT, permeablized with 0.1% Triton in PBS for 15 min at RT, and blocked with the manufacturer's blocking agent (Duolink™ In Situ Red Starter Kit Mouse/Rabbit (Sigma, Cat# DUO92101-1KT) for 1 h followed by incubation with paired primary antibodies: Aurora kinase A (1F8) Mouse mAb (CST 12100; 1:100), Myc Rabbit Ab (CST 13987; 1:100); and GSK-3β Rabbit Ab (27C10) (CST 9315; 1:100). PLA detection was performed in accordance with the accompanied instructions manual. Images were captured by Axio Observer 7 microscope (Zeiss) using 10× objective (0.45 N.A) Image analyses were performed with custom Matlab scripts (ver. R2019b).

**TUNEL and Ki67 staining**. Following rehydration of the paraffin-embedded sections, the slides were exposed to proteinase K (Agilent DAKO) for 5 min at 37 °C. The tissue sections were incubated in TUNEL reaction mixture for 1 h at 37 °C and followed by termination, using the POD solution for 30 min at 37 °C. The chromogen diaminobenzidine was utilized to develop the TUNEL staining, while hematoxylin was employed as a non-specific nuclear stain. Prior to Ki67 staining, slides were incubated in citric acid buffer and heated (antigen retrieval). Thereafter, primary antibody labeling with Ki67 (Dako GA626) was performed for 1.5 h at RT. A secondary antibody horse anti mouse IgG (1:200) was used for 30 min followed by labeling with ABC-Peroxidase solution (1:50) for 30 min at RT.

**Statistics and reproducibility**. Statistical significance was assessed by a two-tailed student's $t$-test or ANOVA (for multiple comparisons) using Prism version 5.04 (GraphPad, La Jolla, CA). A $p \leq 0.05$ was considered statistically significant. CompuSyn software (ComboSyn, Inc., Paramus, NJ) was used for the drug combination analysis including the calculation of the combination index (CI). CI < 1 was considered as synergistic, CI = 1 as an additive, and CI > 1 as antagonistic. All in vitro experiments were performed independently at least twice with similar results.

**Study approval**. All procedures were done in accordance with Animal Welfare Regulations and approved by the Institutional Animal Care and Use Committee at the Columbia University Medical Center (AC-AABC6505 and AC-AAAV7451).

**Reporting summary**. Further information on research design is available in the Nature Research Reporting Summary linked to this article.

## Data availability

The raw and processed ATAC-sequencing, CHIP-sequencing, and microarray data generated in this study have been deposited in the Gene Expression Omnibus (GEO), https://www.ncbi.nlm.nih.gov/geo/, under accession code GSE161572, GSE161573, and GSE152612 respectively. Source data are provided with this paper.

## Code availability

Source code is provided with this paper at https://github.com/scappell/Cell_tracking. Source data are provided with this paper.

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

## Acknowledgements

M.D. Siegelin: NIH NINDS R01NS095848, R01NS102366, R01NS113793, K08NS083732, Louis V. Gerstner, Jr. Scholars Program (2017-2020) and American Brain Tumor Association Discovery Grant 2017 (DG1700013) and Schaefer Research Scholars Program Awards 2020. Trang T.T. Nguyen: American Brain Tumor Association Basic Research Fellowship in Memory of Katie Monson (BRF1900018). Transcriptome analysis was supported by the CTSA grant UL1-TR001430 to the Boston University Microarray and Sequencing Resource Core Facility. These studies used the resources of the Cancer Center Flow Core Facility funded in part through center grant P30CA013696 and S10RR027050. Metabolomics shown in Figs. 2i and 6d–f were performed by the Whitehead Institute Metabolite Profiling Facility (Cambridge, MA). The studies presented in this work were carried out in part in the MRI Facility of the Oncology Precision Therapeutics and Imaging Core (OPTIC) Shared Resource at Columbia University Herbert Irving Comprehensive Cancer Center. The authors wish to thank Dr. Yanping Sun for MRI study design, execution, and assistance in data analysis.

## Author contributions

T.T.T.N. and M.D.S. designed research; T.T.T.N., E.S., C.S., S.K., A.M. (Angeliki Mela), N.H., and A.M. (Aayushi Mahajan) conducted the experiments. T.T.T.N. S.K., H.Y., and M.D.S. analyzed the data. T.T.T.N., M.-A.W., G.K.-M., J.N.B., P.C., and M.D.S. helped with writing, review, and/or revision of the paper. H.O.A., G.Z., and C.M.Q. provided material support. M.D.S. supervised the study.

## Competing interests

The authors declare no competing interests.
