## [Peer Review File · Nature Communications]

Aurora kinase A Inhibition Reverses the Warburg Effect and Elicits Unique Metabolic Vulnerabilities in GlioblastomaREVIEWER COMMENTS

Reviewer #1 (Remarks to the Author):

The study by Nguyen et al. examines the role of Aurora A kinase in GBM growth and metabolism. The authors show that Aurora A kinase regulates protein stability of c-MYC in GBM cell lines, that inhibition of Aurora A kinase results in reversal of the Warburg effect (via downregulation of c-MYC and upregulation of PGC1a), and that combined inhibition of Aurora kinase A and oxidative phosphorylation have additive or synergistic effects on growth inhibition and cell death in GBM cells.

The study is well done and interesting and I have only a few suggestions:

(1) The link of the laboratory data to human GBM datasets is somewhat weak. The analysis of TCGA data for c-MYC and PGC1a (and respective transcriptional programs) should be done at the level of individual tumors and not at the cohort level. Is there RPPA data from human GBM tumors that could perhaps be leveraged to support some of the authors' conclusions?

(2.) It is unclear whether some of the key in-vitro results with SF188 and GBM22 GBM cells are relevant to the observed in-vivo effects. This is particularly important for the proposed key role of c-MYC as (unexpected) effect of Aurora A kinase. The authors therefore should document whether expression of T58A c-MYC also rescues growth inhibition by alisertib in the orthotopic GBM22 model.

(3.) The authors should expand the biochemical analysis of the orthotopic GBM22 tumors to examine phosphorylation of Aurora Kinase A, protein levels and phosphorylation of c-MYC, and RNA levels of some of the key metabolic genes.

(4.) The genotype of the GBM cell lines and PDX models should be described in more detail as GBM cells can harbor many genetic alterations that are immediately relevant to the signaling pathways under examination in this study.

Reviewer #2 (Remarks to the Author):

The manuscript entitled "Aurora A kinase Inhibition Reverses the Warburg Effect and Elicits Unique Metabolic Vulnerabilities in Glioblastoma" describes interesting findings about how AURKA inhibition elicits substantial metabolic reprogramming in Glioblastoma. Overall the results achieved in this study may be of interest to the field and the manuscript represents an amount of work. However, before publication several key issues need to be addressed:

Major points:

1. In Fig1A, why c-myc is picked while it is not the most prominent hit? Among the dots of significant difference, whether there are any enzymes related to glucose/fatty acid metabolism or whether there are classical downstreams of c-myc? Please detailed list of all hits need to be attached.
2. As shown in Fig1B, Why the response concentration for GBM22 is 1000 times higher than SF188? Does it imply that in the vivo assays, a much higher concentration of Alisertib is needed to interfere c-myc expression? Additionally, it is recommended to test the protein expression of c-myc at different time point after treatment with Alisertib, such as 0.5h, 1h, 2h, 4h, 8h.
3. In order to prove that Aurora A and c-myc, Aurora A and GSK3 β have a direct interaction, Co-IP is not enough, the in vitro binding assay and Proximity Ligation Assay are needed.
4. How does Aurora A regulate the phosphorylation of c-myc by GSK3 β in a kinase activity-dependent manner? Does Aurora A simultaneously interact with c-myc and GSK3 β ?
5. It is suggested to detect the changes of interaction between c-myc and Aurora-A, c-myc and GSK3 β after Alisertib treatment.
6. In the discussion, the author should summarize the known functions of Aurora-A in glycolysis and oxidative phosphorylation, and highlight the new findings and significances of this research.
7. please discuss why both knocking down and overexpressing c-myc decrease the sensitivity of

cancer cell to Alisertib?

8. In Fig1B and 1G, after Alisertib treatment, changes of c-myc level in SF188 is inconsistent, what is the difference?

9. In Fig3B and 3C, the baseline is inconsistent. The cells seeded in two groups may be different, and more repeats are required to have reliable results.

Minor points:

1. The subtitle "Aurora A kinase mediated loss of c-Myc facilitates an increase in PGC1 α to drive oxidative metabolism" should be "Aurora A kinase inhibition mediated loss of c-Myc facilitates an increase in PGC1 α to drive oxidative metabolism"

2. In Figure 1A, PGC1 α should also be highlighted ?

3. A working model is necessary to summarize the new findings.

Reviewer #3 (Remarks to the Author):

Summary: Nguyen et al. put forth an elegant paper linking Aurora Kinase inhibition to metabolic pathways. They demonstrate that Aurora Kinase inhibition lowers MYC levels in a GSK3 β dependent manner. They further demonstrate that Aurora Kinase inhibition leads to lowering of HK2 and LDHA levels in a MYC dependent manner, thereby suppressing glycolysis. GBM cells partially resistant to Aurora Kinase inhibition showed increased oxidative mitochondrial metabolism partially mediated by lowering PGC1 α . Aurora Kinase inhibition lowered glucose-derived M+2 citrate and glutamine derived M+4 citrate via oxidative decarboxylation and elevated M+2 citrate from palmitic acid in tracing experiments and that fatty acid oxidation may be elevated in Aurora Kinase resistance. Finally, combination of Aurora Kinase and fatty acid oxidation inhibition led to tumor suppression and prolonging overall survival in 2 independent PDXs. Based on these data, the authors conclude: "Taken together, these data support that simultaneous targeting of oxidative metabolism and AURKAI might be a potential novel therapy against recalcitrant malignancies".

Comments: The authors provide a compelling manuscript that furthers the field by providing a mechanistic link between Aurora Kinase inhibition, resistance and central metabolic pathways. The paper is well-written, clear and beautifully illustrated. The following minor comments are noted to improve clarity of the manuscript.

(1) It is known that MYC expression is a major determinant for antitumor activity of aurora kinase inhibition. SF188 pediatric GBM cells are known to have MYC amplifications. Are these pathways implicated and restricted to cells with known MYC implications? Do the authors know MYC amplification status in GBM22, GBM12, and GBM43 cells?

(2) Figure 5 demonstrates beautiful tracing experiments. A major finding is suppression of lactate production and the Warburg on Aurora Kinase inhibition. To support this observation, can the authors add lactate to their tracing data?

(3) The H&E sections in the combined treatment animals in 7J, 7K, S10E,F,G and S10 H,I,J show very little tumor to compare Ki-67 or Tunnel stains. Can the authors replace these images with sections that show tumor cells? Also, quantification for Ki-67 and Tunnel would help drive the point home.

We are highly appreciative of the constructive critiques provided by the expert reviewers and editors and delighted that we have received the opportunity to resubmit our work to the journal.

We have taken these comments very seriously and have performed additional experiments to improve the paper in accordance with the editors' and reviewers' recommendations.

Figures or panels newly added	
Fig 1m	Animal survival curve of mice implanted GBM22-c-Myc-WT or GBM22-T58A-c-Myc cells in nude mice
Fig 1n	SF188 and GBM22 cells were transfected with non-targeting siNT or siAURKA in the presence or absence of 2 μ M CHIR-908014 for 24h and were subjected to protein capillary electrophoresis for the indicated proteins
Fig 1o	SF188 and GBM22 cells were transfected with HA-EV, HA-Aurora A-WT, HA-Aurora A-D274N and were subjected to protein capillary electrophoresis for the indicated proteins.
Fig 5c	Parental or chronically alisertib treated GBM22 cells were cultured in DMEM media containing 25 mM U- ¹³ C glucose, 4mM glutamine, and 10% dialyzed FBS, and subjected to LC/MS analysis. Shown are relative percentages of the isotopologues for each metabolite (n=3).
Fig S1b	c-Myc protein level of SF188 and GBM22 cells treated with alisertib for 30 min, 1h, 2h, 4h, and 5h with indicated protein.
Fig S1c	c-Myc protein level between SF188 and GBM22 cells.
Fig S1d, e	The tumor size of representative MRI images of vehicle and alisertib treated GBM22 tumors.
Fig S1f,g	Brain tumors from the experiment in (d) were fixed and stained with indicated protein. Quantification is shown in g
Fig S1h,i	The protein capillary electrophoresis of brain tumors treated with Alisertib with indicated protein
Fig S3a	Standard western blot of 293T cells transfected with HA-Aurora A and Flag-Myc for 24h. Cells were lysed and lysates were immunoprecipitated with HA flag bead and probed with indicated antibodies.
Fig S3b, c	Proximity ligation assays analyzing interaction between Myc and Aurora in SF188. Blue is DAPI staining and red is PCR amplification products indicating complex formation of Myc and Aurora A. Quantification of Myc and Aurora interaction was shown in (c)
Fig S3d, e	Proximity ligation assays analyzing interaction between Myc and Aurora in GBM22. Quantification of Myc and Aurora interaction was shown in (e).
Fig S3f, g	SF188 cells were treated with DMSO or alisertib in the presence or absence of 5 μ M of MG132 for 7h. Subsequently, cells lysates were immunoprecipitated with IgG or Myc antibody and probed with indicated antibodies.

Fig S3h, i	Proximity ligation assays analyzing interaction between Myc and Aurora in SF188 treated with DMSO or 1 μ M alisertib. Quantification of Myc and Aurora interaction was shown in (h).
Fig S3l	Standard western blot of Myc protein expression in GBM22 cells infected with EV, Myc-T58A, and Myc-WT.
Fig S4f, g	Proximity ligation assays analyzing interaction between GSK3 β and Aurora in SF188. Blue is DAPI staining and red is PCR amplification products indicating complex formation of GSK3 β and Aurora A. Quantification of GSK3 β and Aurora interaction was shown in (f).
Fig S4h, i	Proximity ligation assays analyzing interaction between GSK3 β and Aurora in GBM22. Quantification of GSK3 β and Aurora interaction was shown in (h).
Fig S5a	Standard western blot of SF188 and GBM22 cells transfected with HA-EV and HA-Aurora A for 24h. Cells were lysed and lysates were immunoprecipitated with HA tag bead and probed with indicated antibodies.
Fig S5b	Standard western blot of 293T cells transfected with Flag-GSK3 β , HA-Myc, and/ or increasing concentration Aurora A (0.25, 0.5, and 1 μ g) in the presence of 5 μ M MG132 for 48h.
Fig S5c	SF188 cells transfected with HA-Aurora A-WT and HA-Aurora A-D274N for 24h and were subjected to protein capillary electrophoresis for the indicated proteins.
Fig S6c	Real time PCR analysis of glycolysis mRNA levels in GBM22 brain tumor treated with vehicle or alisertib (n=5).
Fig S10b	Shown are the correlation of mRNA levels of c-Myc vs PPARGC1A expression in glioblastoma.
Fig S10c	Shown are the correlation of mRNA levels of c-Myc vs PPARGC1A expression in GBM12, GBM22, and GBM43.
Fig S12b	Parental or chronically alisertib treated U87 cells were cultured in DMEM media containing 25 mM U- ¹³ C glucose, 4mM glutamine, and 10% dialyzed FBS, and subjected to LC/MS analysis. Shown are relative percentages of the isotopologues for each metabolite (n=3).
Fig S15h, i	Quantification of TUNEL and Ki67 positive cells in GBM12 and GBM43 tumors treated with vehicle, alisertib, etomoxir, or the combination treatment in experiment in (e) and (h), respectively.
Fig 7l	Quantification of TUNEL and Ki67 positive cells in GBM22 brain tumors treated with vehicle, alisertib, etomoxir, or the combination treatment in experiment in Figure 7h.

Reviewer #1 (Remarks to the Author):

The study by Nguyen et al. examines the role of Aurora kinase A in GBM growth and metabolism. The authors show that Aurora kinase A regulates protein stability of c-MYC in GBM cell lines, that inhibition of Aurora kinase A results in reversal of the Warburg effect (via downregulation of c-MYC and upregulation of PGC1a), and that combined inhibition of Aurora kinase A and oxidative phosphorylation have additive or synergistic effects on growth inhibition and cell death in GBM cells. The study is well done and interesting and I have only a few suggestions:

Response: We thank the reviewer for the appreciation and careful examination of our work.

(1) The link of the laboratory data to human GBM datasets is somewhat weak. The analysis of TCGA data for c-MYC and PGC1a (and respective transcriptional programs) should be done at the level of individual tumors and not at the cohort level. Is there RPPA data from human GBM tumors that could perhaps be leveraged to support some the authors conclusions?

Response: We thank the reviewer for the suggestions and we implement this point in our manuscript in Fig. S10b,c. Based on the cBioportal online platform, we found that GBM PDX lines with high c-Myc mRNA expression will show low PGC1A levels, which is consistent with our TCGA data in Fig. S10a. Unfortunately, there is no RPPA data available for PGC1A. Thus, we cannot correlate the protein levels between c-Myc and PGC1A in the large datasets.

(2) It is unclear whether some of the key in-vitro results with SF188 and GBM22 GBM cells are relevant to the observed in-vivo effects. This is particularly important for the proposed key role of c-MYC as (unexpected) effect of Aurora kinase A. The authors therefore should document whether expression of T58A c-MYC also rescues growth inhibition by alisertib in the orthotopic GBM22 model.

Response: We thank reviewer for the suggestions, and we performed this experiment. We transduced GBM22 cells with lentiviral particles encoding c-Myc T58A and c-Myc wild-type. As

anticipated, host animals carrying orthotopically implanted GBM22 cells expressing the T58A mutated c-Myc were resistant towards alisertib treatment, whereas GBM22 cells carrying wild-type c-Myc were still sensitive to the drug and revealed an extended overall survival. This data has now been implemented as part of Fig.1m and Fig. S3l.

(3) The authors should expand the biochemical analysis of the orthotopic GBM22 tumors to examine phosphorylation of Aurora Kinase A protein levels and phosphorylation of c-MYC, and RNA levels of some of the key metabolic genes.

Response: We thank reviewer for the suggestions, and we collected this data by performing additional experiments. We stereotactically injected GBM22 cells into the right striatum of nude mice (ten animals, five per group). Following randomization, five animals received vehicle and another five were treated with alisertib three times a week. The presence of tumors was confirmed by MRI imaging prior to harvesting the tumors. Thereafter tumors were subjected to protein isolation and the lysates were run on the Wes instrument (protein capillary electrophoresis) to determine the expression levels of c-Myc, which declined following alisertib treatment as in our in vitro results (Figs. S1d-i). Unfortunately, we were not able to detect the levels of phosphorylated c-Myc. As anticipated, we found that alisertib decreased the phosphorylation of Aurora kinase A as determined by immunohistochemistry analysis (Figs. S1f, g). However, we succeeded with the extraction of mRNA and confirmed that akin to the two cell lines (GBM22 and SF188) alisertib suppresses the transcripts of pivotal glycolytic genes (HK2, LDHA and SLC2A1) (Fig. S6c). It is noteworthy that these transcripts are regulated by c-Myc as shown in several instances elsewhere in the manuscript.

(4) The genotype of the GBM cells lines and PDX models should be described in more detail as GBM cells can harbor many genetic alterations that are immediately relevant to the signaling pathways under examination in this study.

Response: As requested, we provided more detail about the GBM lines used in the manuscript (Material and Methods page 25). GBM12 is a PDX line, which are derived from a male patient (glioblastoma, IDH-wildtype, mesenchymal molecular subtype, EGFR amplified, PTEN wt, TP53 mutated (splice mutation), MYC non-amplified). GBM22 is a PDX line, which are derived from a male patient (Gliosarcoma, IDH-wildtype, classical molecular subtype, IDH1 wt, EGFR gain, PTEN wt, TP53 mutated, MYC amplified). GBM43 is a PDX line, which is derived from a male patient (Glioblastoma, IDH-wildtype, classical molecular subtype, IDH wt, EGFR gain, PTEN wildtype, TP53 mutated, MYC non-amplified). SF188 is a pediatric glioblastoma cell line derived from 8-year old male (TP53 mut, PTEN wt, MYC-amplified)

Reviewer #2 (Remarks to the Author):

The manuscript entitled “Aurora kinase A Inhibition Reverses the Warburg Effect and Elicits Unique Metabolic Vulnerabilities in Glioblastoma” describes interesting findings about how AURKA inhibition elicits substantial metabolic reprogramming in Glioblastoma. Overall, the results achieved in this study may be of interest to the field and the manuscript represents an amount of work. However, before publication several key issues need to be addressed:

Response: We thank the reviewer for the appreciation and careful examination of our work.

Major points:

1. In Fig1A, why c-myc is picked while it is not the most prominent hit? Among the dots of significant difference, whether there are any enzymes related to glucose/fatty acid metabolism or whether there are classical downstreams of c-myc? Please detailed list of all hits need to be attached.

Response: We fully agree that this is an important point. We chose c-Myc as the most prominent hit because of its relevance to our study. SF188 and GBM22 are cell cultures that display high c-Myc levels and in turn critically rely on it for their survival and proliferation (1). For this reason, the

displayed change in c-Myc protein level upon alisertib treatment measured by RPPA analysis is significant. In addition, c-Myc is a known master regulator of cell proliferation, cell death, and metabolism. Therefore, changes in its levels will likely have a substantial effect, especially in the setting of high c-Myc expressing cells. The complete list of hits has been attached as Table S2.

2. As shown in Fig1B, Why the response concentration for GBM22 is 1000 times higher than SF188? Does it imply that in the vivo assays, a much higher concentration of Alisertib is needed to interfere c-myc expression? Additionally, it is recommended to test the protein expression of c-myc at different time point after treatment with Alisertib, such as 0.5h, 1h, 2h, 4h, 8h.

Response: According to the online platform (cBioportal), both SF188 and GBM22 express high levels of c-Myc mRNA expression; however, the cMyc expression in these cell lines are still different. SF188 reveals higher c-Myc protein level as compared to GBM22, which in part may explain the difference in susceptibility to alisertib treatment. To highlight this difference, we analyzed the expression levels of c-Myc by protein capillary electrophoresis and included this data as part of Fig. S1c. Regarding the in vivo experiments, we administered the same dosage of Alisertib for all GBM models (GBM22, GBM12, and GBM43) and they all appear to respond in a similar fashion. It should be noted that SF188 cells are non-tumorigenic and therefore could not be tested in vivo. As recommended by the reviewer, we confirmed the expression of c-Myc protein in these cell lines at different time points as suggested. This data has been included as part of Fig. S1b

3. In order to prove that Aurora A and c-myc, Aurora A and GSK3 β have a direct interaction, Co-IP is not enough, the in vitro binding assay and Proximity Ligation Assay are needed.

Response: As requested, we performed an additional technique to prove the interactions between the proteins. We utilized the proximity ligation assay to demonstrate that c-Myc and Aurora A or Aurora A and GSK3 β interact (Figs. S3b-e and Fig. S4f-i). Moreover, by using this assay we

showed that the interaction between c-Myc and Aurora A are disrupted (Fig. S3h, i). Moreover, we have performed additional Co-IP analysis with ectopic HA-tagged Aurora kinase A and Flag-tagged c-Myc to further validate the interaction (Fig. S3a). Previously, one other group confirmed that GSK3B and Aurora kinase A directly interact (2.). Similarly, a recent paper showed that recombinant Aurora kinase A interacts with c-MYC peptides directly, proving the in vitro interactions (3).

4. How does Aurora A regulate the phosphorylation of c-myc by GSK3 β in a kinase activity-dependent manner? Does Aurora A simultaneously interact with c-myc and GSK3 β ?

Response: This is an exciting question and we performed additional experiments to provide answers. First, we demonstrated through an additional co-IP by pulling down Aurora kinase A that Aurora kinase A, GSK3B and c-Myc interact simultaneously in GBM22 and SF188 cells (Fig S5a). To further clarify this interaction and the role of Aurora kinase A we performed an additional co-IP analysis in which three plasmids (untagged-Aurora A, Flag-GSK3B, and HA-c-MYC) were simultaneously transfected. The Aurora-A plasmid was used at three different dosages to demonstrate that Aurora kinase A disrupts the interaction between c-Myc and GSK3B and thereby inhibits the phosphorylation of c-Myc at T58 by GSK3B (Fig. S5b). Indeed, and as expected, we found that Aurora kinase A blocks the interaction of c-Myc and GSK3B, while at the same time the interaction between Aurora kinase A and GSK3B was relatively maintained/preserved (Figs. S3f, g). These findings further validate that the three proteins are present in a complex and that Aurora kinase A appears to safe guard c-Myc from its GSK3B dependent degradation through the proteasome. Next, we extended our data related to the interaction and phosphorylation of GSK3B by Aurora kinase A. In addition, we have demonstrated that loss of Aurora kinase A by genetic interference or pharmacological blockage of its kinase activity (through alisertib) leads to a reduction of c-Myc dependent on GSK3B kinase activity by pharmacological inhibitors of GSK3B and through genetic silencing (Fig 1n and Figs. S4a-d). We have further extended this data to

demonstrate that inhibition of GSK3B potentially rescued c-Myc protein levels and reversed T58 phosphorylation of c-Myc following Aurora kinase A inhibition (Fig. 1n). It should be noted that it was shown earlier by others that Aurora kinase A directly phosphorylates GSK3B at serine 9 in the context of an in vitro kinase assay (2). To extend these findings related to MYC phosphorylation/degradation further in the context of glioblastoma cells, we have generated a kinase dead mutant of Aurora kinase A (D274N). When compared to wild-type Aurora kinase A the kinase dead mutant revealed reduced ability to phosphorylate GSK3B at serine 9 (Fig. S5c). Moreover, only wild-type Aurora kinase A increases c-Myc levels (Figure 1o). Thus, these observations support the idea that Aurora kinase A regulates the phosphorylation of c-Myc by GSK3B in a kinase dependent manner.

5. It is suggested to detect the changes of interaction between c-myc and Aurora-A, c-myc and GSK3 β after Alisertib treatment.

Response: As requested, we performed co-IP to detect the changes of the interaction between c-Myc and Aurora kinase A upon Alisertib treatment in GBM cells. We observed a disruption of the binding between Aurora A and c-Myc and between c-Myc and GSK3 β (Figs S3f, g). Comparable findings were seen in the proximity ligation assay (Figs. S3h, i). In addition, we found that following silencing of Aurora kinase A, GSK3B is less phosphorylated which led to the activation of GSK3B activity resulting in enhanced phosphorylation of c-Myc at T58 (Fig. 1n).

6. In the discussion, the author should summarize the known functions of Aurora-A in glycolysis and oxidative phosphorylation, and highlight the new findings and significances of this research.

Response: We thank the reviewer for this suggestion and we implemented it in the discussion part (Manuscript page 24).

7. please discuss why both knocking down and overexpressing c-myc decrease the sensitivity of cancer cell to Alisertib?

Response: We thank the reviewer for this important question. As previously shown by Otto et al. in another context (MYCN and AURKA) (4), modest over-expression of MYCN is able to rescue from AURKA inhibition mediated reduction in neuroblastoma viability. Our findings are in line with this observation since we also noted a modest rescue from alisertib mediated reduction of cellular viability in two GBM models. Regarding silencing, it is anticipated that when the primary target/effector of a drug is removed the efficacy will be expected to decrease. We have added a small section to the discussion and included the new reference (Manuscript page 23).

8. In Fig1B and 1G, after Alisertib treatment, changes of c-myc level in SF188 is inconsistent, what is the difference?

Response: While the degree of c-Myc protein level suppression may be slightly different (most likely due to experimental variabilities in the indicated experiments), the net results is that c-Myc protein is significantly down-regulated by alisertib in the cell lines tested. Moreover, to provide additional evidence that alisertib suppresses c-Myc protein levels we performed a time course experiment to confirm the down regulation of c-Myc upon the alisertib treatment in two GBM models (Fig. S1b).

9. In Fig3B and 3C, the baseline is inconsistent. The cells seeded in two groups may be different, and more repeats are required to have reliable results.

Response: We thank the reviewer for this important remark. The Seahorse data showed a substantial increase of the oxygen consumption rate (OCR) in alisertib exposed cells (chronic treatment) in Fig. 3b (SF188) and Fig. 3c (GBM22), which is related to the metabolic reprogramming elicited by the drug treatment and which we have genetically confirmed as well.

In our opinion, a different OCR baseline between the two cell lines Fig. 3b (SF188) and Fig. 3c (GBM22) is very much expected given the different phenotype of these cells.

Minor points:

1. The subtitle “Aurora kinase A mediated loss of c-Myc facilitates an increase in PGC1 α to drive oxidative metabolism ”should be “Aurora kinase A inhibition mediated loss of c-Myc facilitates an increase in PGC1 α to drive oxidative metabolism”

Response: We agree with the reviewer and have changed the subtitle in the manuscript (page 13).

2. In Figure 1A, PGC1 α should also be highlighted?

Response: We thank the reviewer for raising this critical point. For our initial screening, we performed the RPPA analysis with the pre-offered/standard 151 antibodies in Alisertib treated SF188. We observed a substantial down regulation of the c-Myc protein, a master regulator of cell proliferation and metabolism. Later on, by performing a transcriptome analysis, we found that following alisertib exposure GBM cells activate PPARA pathways (with increases in PGC1 α) and oxidative metabolism accompanied by inhibition of c-MYC signaling. For more information about the 151 antibodies tested in the volcano graph, please refer to the link: <https://www.mdanderson.org/research/research-resources/core-facilities/functional-proteomics-rppa-core/antibody-information-and-protocols.html>.

3. A working model is necessary to summary the new findings.

Response: As requested, we have a graphical abstract for the findings in this project (Graphical Abstract).

Reviewer #3 (Remarks to the Author):

Summary: Nguyen et al. put forth an elegant paper linking Aurora Kinase inhibition to metabolic pathways. They demonstrate that Aurora Kinase inhibition lowers MYC levels in a GSK3 β dependent manner. They further demonstrate that Aurora Kinase inhibition leads to lowering of HK2 and LDHA levels in a MYC dependent manner, thereby suppressing glycolysis. GBM cells partially resistant to Aurora Kinase inhibition showed increased oxidative mitochondrial metabolism partially mediated by lowering PGC1 α . Aurora Kinase inhibition lowered glucose-derived M+2 citrate and glutamine derived M+4 citrate via oxidative decarboxylation and elevated M+2 citrate from palmitic acid in tracing experiments and that fatty acid oxidation may be elevated in Aurora Kinase resistance. Finally, combination of Aurora Kinase and fatty acid oxidation inhibition led to tumor suppression and prolonging overall survival in 2 independent PDXs. Based on these data, the authors conclude: "Taken together, these data support that simultaneous targeting of oxidative metabolism and AURK*Ai* might be a potential novel therapy against recalcitrant malignancies".

Comments: The authors provide a compelling manuscript that furthers the field by providing a mechanistic link between Aurora Kinase inhibition, resistance and central metabolic pathways. The paper is well-written, clear and beautifully illustrated. The following minor comments are noted to improve clarity of the manuscript.

Response: We thank the reviewer for the appreciation and careful examination of our work.

(1) It is known that MYC expression is a major determinant for antitumor activity of aurora kinase inhibition. SF188 pediatric GBM cells are known to have MYC amplifications. Are these pathways implicated and restricted to cells with known MYC implications? Do the authors know MYC amplification status in GBM22, GBM12, and GBM43 cells?

Response: Based on the online portal cBioportal, MYC amplification is found in GBM22, but not in GBM12 and GBM43. However, c-Myc protein is still expressed in these lines (Material and Method, page 25). Therefore, our studied pathway is not restricted to cell lines that harbor MYC

amplification, it only needs the expression of c-Myc. In GBM, MYC amplification is found in a small percentage of cases. However, c-Myc still plays a central role in cell proliferation and metabolism since c-Myc levels are not only controlled at the transcriptional level (regulated through DNA amplification of the MYC locus), but rather through posttranslational regulation, e.g. phosphorylation and subsequent proteasomal degradation. Because GBM often shows alterations in receptor kinase signaling, e.g. they harbor EGFRvIII mutations, EGFR amplification, loss of PTEN, the ultimate consequence is substantial activation of Akt and mTOR signaling. It is well known that Akt phosphorylates GSK3B at serine 9 to inactivate it. In turn, c-Myc is less phosphorylated at T58 by GSK3B, becomes stabilized and more abundant in these cells, which is summarized in (5). Similarly, c-Myc is increased and plays a major role in GBM stem cells, while not being necessarily amplified at the DNA level. In this vein, both GBM12 and GBM43 have known deregulated receptor kinase pathways and thereby promote c-Myc stability and activity.

(2) Figure 5 demonstrates beautiful tracing experiments. A major finding is suppression of lactate production and the Warburg on Aurora Kinase inhibition. To support this observation, can the authors add lactate to their tracing data?

Response: We thank the reviewer for the suggestion and implement the requested data in Fig. 5c, Fig. S12b and in Manuscript result part (page 16). Cancer cells consume ample amounts of glucose and metabolize it via pyruvic acid to lactic acid, a phenomenon, which is also referred as the Warburg effect. Our U-¹³C-glucose tracing showed a significant decrease of the m+3 isotopologue of both lactate and pyruvate in Alisertib treated GBM22 cells. Consistently, unlabeled pyruvate and lactate (m+0 pyruvate, m+0 lactate) are increased following alisertib exposure, in keeping with decreased labeling of these metabolites by U-¹³C-glucose.

(3) The H&E sections in the combined treatment animals in 7J, 7K, S10E,F,G and S10 H,I,J show very little tumor to compare Ki-67 or Tunnel stains. Can the authors replace these images with

sections that show tumor cells? Also, quantification for Ki-67 and Tunnel would help drive the point home.

Response: As requested, we provide an H&E section that shows more tumor cells (Fig S15e, and Fig. S15j). The quantification of Ki67 and TUNEL were analyzed to demonstrate a significant difference between single treatment and combination treatment (Fig. S15h, I and Fig. 7I). For figure 7i, we felt that the actual representation through the H&E is representative of the treatment, which results in tumor cell necrosis bordered by viable tumor.

References:

1. Wise DR, DeBerardinis RJ, Mancuso A, Sayed N, Zhang XY, Pfeiffer HK, Nissim I, Daikhin E, Yudkoff M, McMahon SB, Thompson CB. Myc regulates a transcriptional program that stimulates mitochondrial glutaminolysis and leads to glutamine addiction. *Proc Natl Acad Sci U S A*. 2008;105(48):18782-7. Epub 2008/11/27. doi: 10.1073/pnas.0810199105. PubMed PMID: 19033189; PMCID: PMC2596212.
2. Dar AA, Belkhiri A, El-Rifai W. The aurora kinase A regulates GSK-3beta in gastric cancer cells. *Oncogene*. 2009;28(6):866-75. Epub 2008/12/09. doi: 10.1038/onc.2008.434. PubMed PMID: 19060929; PMCID: PMC2642527.
3. Dauch D, Rudalska R, Cossa G, Nault JC, Kang TW, Wuestefeld T, Hohmeyer A, Imbeaud S, Yevsa T, Hoenicke L, Pantzar T, Bozko P, Malek NP, Longerich T, Laufer S, Poso A, Zucman-Rossi J, Eilers M, Zender L. A MYC-aurora kinase A protein complex represents an actionable drug target in p53-altered liver cancer. *Nat Med*. 2016;22(7):744-53. Epub 2016/05/24. doi: 10.1038/nm.4107. PubMed PMID: 27213815.
4. Otto T, Horn S, Brockmann M, Eilers U, Schuttrumpf L, Popov N, Kenney AM, Schulte JH, Beijersbergen R, Christiansen H, Berwanger B, Eilers M. Stabilization of N-Myc is a critical function of Aurora A in human neuroblastoma. *Cancer Cell*. 2009;15(1):67-78. Epub 2008/12/30. doi: 10.1016/j.ccr.2008.12.005. PubMed PMID: 19111882.
5. Bi J, Chowdhry S, Wu S, Zhang W, Masui K, Mischel PS. Altered cellular metabolism in gliomas - an emerging landscape of actionable co-dependency targets. *Nat Rev Cancer*. 2020;20(1):57-70. Epub 2019/12/07. doi: 10.1038/s41568-019-0226-5. PubMed PMID: 31806884.

REVIEWER COMMENTS

Reviewer #1 (Remarks to the Author):

The authors addressed my concerns. The additional data and related manuscript revisions considerably strengthened the manuscript.

Reviewer #2 (Remarks to the Author):

The manuscript has displayed a drastic improvement. All my concerns have been adequately addressed.

Reviewer #3 (Remarks to the Author):

The authors have addressed all comments.

Reviewer #1 (Remarks to the Author):

The authors addressed my concerns. The additional data and related manuscript revisions considerably strengthened the manuscript.

Response: We thank the reviewer for the careful examination of our work and are delighted that she/he recommended our paper for publication.

Reviewer #2 (Remarks to the Author):

The manuscript has displayed a drastic improvement. All my concerns have been adequately addressed.

Response: Thank you very much for your support and improvement of our work.

Reviewer #3 (Remarks to the Author):

The authors have addressed all comments.

Response: We thank this reviewer for all his valuable comments.